Journal of Data-centric Machine Learning Research (2025)        Submitted 9/24; Revised 1/25; Published 2/25

# SuperBench: A Super-Resolution Benchmark Dataset for Scientific Machine Learning

**Pu Ren**[1,*]                                                                     PREN@LBL.GOV
**N. Benjamin Erichson**[1,2,*]                                        ERICHSON@LBL.GOV
**Junyi Guo**[2]                                                     JUNYI@ICSI.BERKELEY.EDU
**Shashank Subramanian**[1]                      SHASHANKSUBRAMANIAN@LBL.GOV
**Omer San**[3]                                                                    OSAN@UTK.EDU
**Zarija Lukić**[1]                                                              ZARIJA@LBL.GOV
**Michael W. Mahoney**[1,2,4]                      MMAHONEY@STAT.BERKELEY.EDU

[1] *Lawrence Berkeley National Lab*
[2] *International Computer Science Institute*
[3] *University of Tennessee, Knoxville*
[4] *University of California at Berkeley*
∗ *Equal contribution.*

**Reviewed on OpenReview:** *https: // openreview. net/ forum? id= 0J6zUcWldW*

**Editor:** Holger Caesar

## Abstract

Super-resolution (SR) techniques aim to enhance data resolution, enabling the retrieval of finer details, and improving the overall quality and fidelity of the data representation. There is growing interest in applying SR methods to complex spatiotemporal systems within the Scientific Machine Learning (SciML) community, with the hope of accelerating numerical simulations and/or improving forecasts in weather, climate, and related areas. However, the lack of standardized benchmark datasets for comparing and validating SR methods hinders progress and adoption in SciML. To address this, we introduce SuperBench (https://github.com/erichson/SuperBench), the first benchmark dataset featuring high-resolution datasets (up to $2048 \times 2048$ dimensions), including data from fluid flows, cosmology, and weather. Here, we focus on validating spatial SR performance from data-centric and physics-preserved perspectives, as well as assessing robustness to data degradation tasks. While deep learning-based SR methods (developed in the computer vision community) excel on certain tasks, despite relatively limited prior physics information, we identify limitations of these methods in accurately capturing intricate fine-scale features and preserving fundamental physical properties and constraints in scientific data. These shortcomings highlight the importance and subtlety of incorporating domain knowledge into ML models. We anticipate that SuperBench will help to advance SR methods for science.

**Keywords:**   super-resolution, scientific machine learning, benchmark dataset

## 1 Introduction

Super-resolution (SR) techniques have emerged as powerful tools for enhancing data resolution, improving the overall quality and fidelity of data representation, and retrieving fine-scale structures. These techniques find application in diverse fields such as image restoration and

enhancement (Park et al., 2003; Van Ouwerkerk, 2006; Tian and Ma, 2011; Nasrollahi and Moeslund, 2014), medical imaging (Greenspan, 2009; Isaac and Kulkarni, 2015), astronomical imaging (Puschmann and Kneer, 2005; Li et al., 2018), remote sensing (Arefin et al., 2020), and forensics (Satiro et al., 2015). Despite the remarkable achievements of deep learning-based SR methods developed primarily in the computer vision community (Anwar et al., 2020; Van Ouwerkerk, 2006), their application to scientific tasks has certain limitations. In particular, these methods, state-of-the-art (SOTA) within machine learning (ML), often struggle to capture intricate fine-scale features accurately and to preserve fundamental physical properties and constraints inherent in scientific data. These shortcomings highlight the challenge of incorporating domain knowledge into ML models. While these are well-known anecdotal observations (that we confirm), a more basic issue is that the progress and widespread adoption of SR methods in scientific machine learning (SciML) community face a significant challenge: the absence of standardized benchmark datasets for comparing and validating the performance of different SR approaches. To address this crucial gap, we introduce `SuperBench`, an innovative benchmark dataset that fills the need for standardized evaluation and comparison of SR methods within scientific domains.

**Problem setup.** SR is a task that involves recovering fine-scale data from corresponding coarse-grained data. In the context of scientific SR, let us consider the example of weather data that captures complex interactions among the atmosphere, oceans, and land surface, illustrated in Figure 1. The coarse-grained data can be regarded as a down-sampled version of the fine-scale data. The former coarse-scale data can be represented as low-dimensional data $x \in \mathcal{X}$; while the latter fine-scale data can be seen as high-dimensional data $y \in \mathcal{Y}$.

In practice, there are various degradation functions $f$ that can generate the coarse-grained data. We can model the degradation process as $x = f(y) + \epsilon$, where $f : \mathcal{Y} \to \mathcal{X}$ is a degradation function, and $\epsilon$ represents noise. The degradation function $f$ can be non-linear, and the noise term $\epsilon$ can have complex spatial and temporal patterns. For example, a simple degradation function commonly used is bicubic down-sampling (Anwar et al., 2020; Van Ouwerkerk, 2006). However, this simplistic approach does not capture the challenges of real-world SR problems (Cai et al., 2019; Lugmayr et al., 2019), where more complex unknown degradations are typically encountered. Thus, we also consider more realistic degradations, such as uniform down-sampling with noise, to simulate experimental measurement setups, as well as direct low-resolution (LR) simulations of data. These scenarios pose significant challenges for SR techniques. SR works in the opposite direction of down-sampling, aiming to recover a high-resolution (HR) representation (including fine-scale structures) from the given coarse-grained data $x$. More concretely, the aim is to find an inverse map $f^{-1} : \mathcal{X} \to \mathcal{Y}$ that accurately restores the fine-scale details. SR is inherently difficult due to the complex fine-scale structures in high-dimensional data, which cannot be fully described by the limited information available in low-dimensional data. This inverse problem often has multiple solutions, especially when dealing with higher up-sampling factors from the low-dimensional to high-dimensional space.

In this paper, we are concerned with establishing a high-quality benchmark dataset, `SuperBench`, for spatial SR methods for scientific problems. By providing a standardized benchmark dataset, `SuperBench` empowers SciML researchers to evaluate and advance SR methods

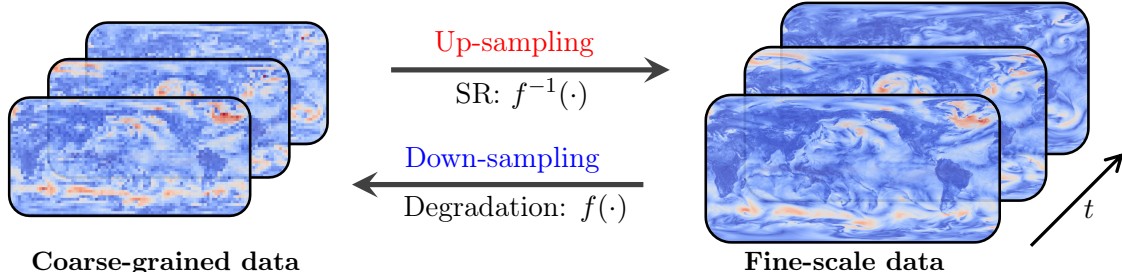

Figure 1: High-resolution data are paramount to accurately resolving the turbulent dynamics of Earth's weather systems. For instance, resolving storms requires kilometer-scale resolutions, and some crucial climate processes can require order of 1m resolutions. The snapshots on the left show coarse-grained data that can be thought of as a down-sampled representation of the fine-scale data on the right. Coarse-grained data not only fail to capture the small scales, but they also do not account for the impact of these small scales on the large-scale dynamics, nor the impact of fine (and critical) topographic features such as mountain ranges on either scale. Currently, generating these high-resolution and accurate data demands prohibitive computational resources (thousands of nodes on modern super-computing substrates). Based on current computing trends, it may be several decades before numerical solvers of atmospheric physics can simulate at a meter resolution (Schneider et al., 2017), which represents a grand challenge to scientific computing. Using SR to resolve fine-scale structures from coarser simulations holds an enormous promise towards fast, efficient, and accurate models for atmospheric physics emulation.

specifically tailored for scientific tasks. We anticipate that this benchmark dataset will significantly contribute to the advancement of SR techniques in scientific domains, fostering the development of more effective and reliable methods for enhancing data resolution and improving scientific insights.

**Main contributions.** The key contributions of this paper are summarized as follows.

- We introduce `SuperBench` (https://github.com/erichson/SuperBench), a novel benchmark dataset comprising high-quality scientific data for spatial SR methods. This dataset includes four distinct datasets of HR simulations, with dimensions up to $2048 \times 2048$, surpassing the resolution of typical scientific datasets used in SciML. `SuperBench` has a total file size of 439 GB. The datasets feature challenging problems in fluid flows, cosmology, and climate science. They are specifically chosen to push the performance limits of existing methods and facilitate the development of innovative SR methods for scientific applications.

- We investigate a range of degradation functions tailored for scientific data. In addition to commonly used methods like uniform and bicubic downscaling, we explore the use of LR simulations as inputs and consider the introduction of noise to the input data. This suits `SuperBench` for a thorough assessment and effective comparison of different SR methods.

- We benchmark existing SR methods on `SuperBench`. By employing both data and physical-centric metrics, our analysis provides valuable insights into the performance of various SR approaches. Notably, our findings demonstrate that purely data-driven SR methods, even those employing advanced architectures like Transformers, struggle to preserve the

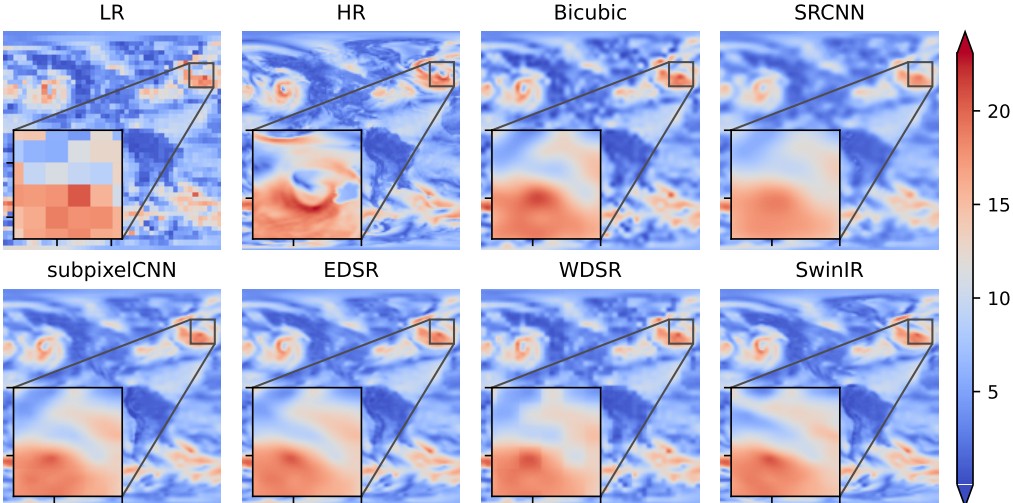

Figure 2: A cropped example snapshot of weather data. The task is to recover the HR representation from the corresponding LR input by a factor of ×16. All SOTA methods reconstruct a blurred approximation that washes out important multi-scale and fine-scale features of physical importance.

physical properties of turbulence datasets. An illustrative example of the performance of baseline models on weather data is shown in Figure 2, which shows the limitations of current approaches for modeling multi-scale structures.

The motivation behind creating an SR benchmark dataset for scientific problems stems from two key factors: (i) the prohibitively high computational cost associated with executing HR numerical simulations; and (ii) the inherent limitations of measurements in large-scale experiments, which often have restricted resolution. It is important to note that SR applied to scientific data differs from its application to general image data in two significant ways. Firstly, many physical systems adhere to explicit governing laws and exhibit distinct features at fine scales, such as multi-scale turbulence phenomena. Thus, preserving the inherent physical properties of scientific data during the SR process becomes a crucial objective. As a potential research direction, exploring constrained ML methods, including soft versus hard constraints and equality versus inequality constraints, could prove fruitful here (Krishnapriyan et al., 2021; Edwards, 2022; Négiar et al., 2022). Secondly, the evaluation metrics for SR on scientific data may differ, as scientists are primarily concerned with pixel-wise reconstruction accuracy and specific domain-dependent metrics. For these and other reasons, assessing the performance of SR methods necessitates a high-quality scientific benchmark dataset.

**Limitations.** We limit the scope of this initial benchmark dataset to spatial SR tasks, which still pose a range of challenges for existing SR methodology in the context of scientific applications. However, we note that there is an increasing interest in applying SR methods to dynamical system applications (e.g., videos or fluid flow) where the model aims to recover temporal or both spatial and temporal information.

## 2 Related Work

A wide range of methods exists for SR (Nasrollahi and Moeslund, 2014). However, it has been well-established that deep learning offers a powerful and versatile framework for SR solutions, as demonstrated in (Anwar et al., 2020), (Wang et al., 2020c), and the references therein. Moreover, recent studies have shown the effectiveness of deep learning-based SR methods specifically for fluid flows (Fukami et al., 2019; Liu et al., 2020; Erichson et al., 2020; Bao et al., 2022; Fukami et al., 2023). In the following, we provide a brief overview of the most notable deep learning-based SR methods that are relevant to our benchmark dataset.

**Single-image SR methods.** Single-image SR (SISR) focuses on tackling spatial SR. SRCNN (Dong et al., 2015) is the first iconic work of introducing deep convolutional neural networks (CNNs) for SISR; and it drastically improves the reconstruction performance, compared with the traditional method based on sparse representation (Yang et al., 2010). In addition to the interpolation-based up-sampling used in SRCNN, the strategies of deconvolution (Dong et al., 2016) and pixelshuffle (Shi et al., 2016) also attract considerable critical attention. Furthermore, due to a proliferation of network designs, we observe increasingly rapid advances in the field of SISR. Generally, there are six model architectures: residual networks (Kim et al., 2016; Lim et al., 2017; Yu et al., 2018); recursive blocks (Tai et al., 2017); dense networks (Tong et al., 2017; Zhang et al., 2018); generative adversarial networks (GAN) (Ledig et al., 2017; Wang et al., 2018; Zhang et al., 2019; Chan et al., 2021); attention schemes (Yang et al., 2020; Chen et al., 2021; Liang et al., 2021); and diffusion models (Rombach et al., 2022; Saharia et al., 2022).

**Constrained SR.** Recently, researchers have shown great interest in SR of scientific data (e.g., fluid flow). For example, neural networks have been introduced to reconstruct fluid flows by learning an end-to-end mapping between LR data and HR solution field based on either limited sensor measurements (Erichson et al., 2020) or sufficient labeled simulations (Xie et al., 2018; Yu and Hesthaven, 2019; Fukami et al., 2019; Liu et al., 2020; Fukami et al., 2021a,b; Vinuesa and Brunton, 2022; Fukami et al., 2023). Moreover, many scientists have started to explore the potential of incorporating domain-specific constraints into the learning process, due to the accessibility of physical principles. In the context of scientific tasks, the realm of constrained SR has been investigated in two primary directions. The first direction involves the integration of constraints into the loss function, which guides the optimization process. In specific, the popular physics-informed neural networks (PINNs) (Raissi et al., 2019; Karniadakis et al., 2021; Krishnapriyan et al., 2021; Edwards, 2022) are essentially constructed in a soft-constraint strategy. Due to the simplicity of this penalty method, there have been various downstream applications of physics-informed SR for scientific data (Wang et al., 2020a; Subramaniam et al., 2020; Gao et al., 2021; Esmaeilzadeh et al., 2020; Ren et al., 2023; Bode et al., 2021).

**Related benchmarks.** In recent years, a number of benchmarks and datasets have been developed to facilitate the evaluation and comparison of SR methods. Among them, several prominent benchmarks have gained wide recognition as standardized evaluation datasets for both traditional and deep learning-based SR methods, such as Train91 (Yang et al., 2008), Set5 (Bevilacqua et al., 2012), Set14 (Zeyde et al., 2010), B100 (Martin et al., 2001), Urban100 (Huang et al., 2015), and 2K resolution high-quality DIV2K (Agustsson and

Timofte, 2017). Moreover, the bicubic down-sampling is the most commonly employed degradation operator to simulate the transformation from HR to LR images.

In addition, specialized benchmarks and datasets have been developed to address the distinct challenges and specific requirements of scientific applications. For example, PDEBench (Takamoto et al., 2022) serves as a benchmark suite for a wide range of Partial Differential Equations (PDEs) simulation tasks. Recently, there has been an emergence of scientific datasets for machine learning research from various domains, including environmental science (Yeh et al., 2021), climate science (Yu et al., 2024), turbulence flows (Chung et al., 2024), and multiphase multiphysics (Hassan et al., 2024). Furthermore, the open-source library DeepXDE (Lu et al., 2021b) offers comprehensive scientific ML solutions, particularly focusing on PINN (Raissi et al., 2019) and DeepONet (Lu et al., 2021a) methods.

## 3 Description of `SuperBench`

`SuperBench` serves as a benchmark dataset for evaluating spatial SR methods in scientific applications. It aims to achieve two primary objectives: (1) expand the currently available SR datasets, particularly by incorporating HR datasets with dimensions such as $2048 \times 2048$ and beyond; and (2) enhance the diversity of data by extending the scope of SR to scientific domains. To achieve these goals, we specifically focus on fluid flows, cosmology, and weather data. These data exhibit multi-scale structures that present challenging problems for SR methods. Examples are shown in Figure 3.

### 3.1 Datasets

Table 1 presents a brief summary of the datasets included in `SuperBench`. The benchmark dataset comprises four different datasets, from three different scientific domains. Among them are two fluid flow datasets, featuring varying Reynolds numbers ($Re$) to capture different flow regimes. Additionally, `SuperBench` includes a cosmology dataset and a weather dataset, ensuring a diverse range of scientific contexts for evaluating SR methods. Note that all the experiments in `SuperBench` can be reproduced on an NVIDIA A100 GPU with 40GB memory, which ensures accessibility for researchers with standard computational resources.

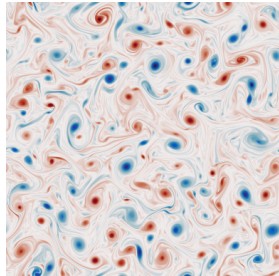 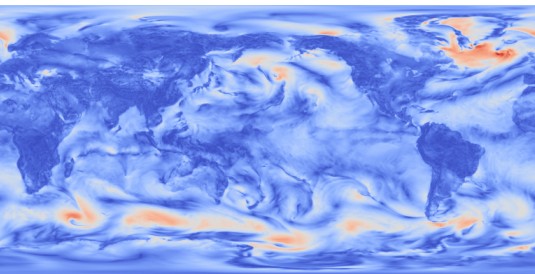 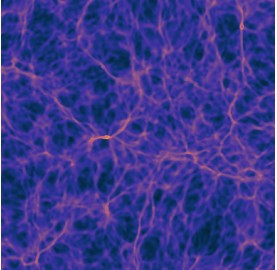

Figure 3: High-resolution example snapshots included in `SuperBench`, showing a Navier-Stokes Kraichnan Turbulence fluid flow (left), weather data that are comprised of several atmospheric variables (middle), and simulated cosmology hydrodynamics data (right).

Table 1: Summary of datasets in `SuperBench`. "LR sim." denotes that the LR simulation data is included in this dataset as inputs.

| Datasets | Spatial resolution | # samples (train/valid/test) | File size |
|---|---|---|---|
| Fluid flow data ($Re = 16000$) | $2048 \times 2048$ | 1000 / 200 / 200 | 66GB |
| $\hookrightarrow$ (w/ LR sim.) | $2048 \times 2048$ | 1200 / 200 / 200 | 80GB |
| Fluid flow data ($Re = 32000$) | $2048 \times 2048$ | 1000 / 200 / 200 | 66GB |
| $\hookrightarrow$ (w/ LR sim.) | $2048 \times 2048$ | 1200 / 200 / 200 | 80GB |
| Cosmology data | $2048 \times 2048$ | 1000 / 200 / 200 | 44GB |
| $\hookrightarrow$ (w/ LR sim.) | $2048 \times 2048$ | 1200 / 200 / 100 | 64GB |
| Weather data | $720 \times 1440$ | 1460 / 365 / 730 | 39GB |

### 3.1.1 NAVIER-STOKES KRAICHNAN TURBULENCE (NSKT) FLUID FLOWS

Fluid flows are ubiquitous in diverse scientific, engineering, and technological domains, including environmental science, material sciences, geophysics, astrophysics, and chemical engineering; and the understanding and analysis of fluid flows have significant implications across these disciplines. In particular, turbulence is a chaotic phenomenon that arises within fluid flows. While the Navier-Stokes (NS) equations serve as a fundamental framework for elucidating fluid motion, their solution becomes increasingly arduous and challenging in the presence of turbulence. The NS equations that couple the velocity field to pressure gradients are given by

$$\nabla \cdot \mathbf{u} = 0, \quad \frac{\partial \mathbf{u}}{\partial t} + \mathbf{u} \cdot \nabla \mathbf{u} = -\frac{1}{\rho}\nabla \mathbf{p} + \nu \nabla^2 \mathbf{u}, \tag{1}$$

where $\mathbf{u}$ is the velocity field and $\mathbf{p}$ is the pressure. Moreover, $\rho$ and $\nu$ denote the density and the viscosity, respectively. The Kraichnan model provides a simplified approach to studying turbulent and chaotic behavior in fluids. In this work, we consider two-dimensional Kraichnan turbulence in a doubly periodic square domain within $[0, 2\pi]^2$ (Pawar et al., 2023). The spatial domain is discretized using $2048^2$ degrees of freedom (DoF), and the solution variables of NS equations are obtained from direct numerical simulation (DNS). A second-order energy-conserving Arakawa scheme (Arakawa, 1997) is employed for computing the nonlinear Jacobian, and a second-order finite-difference scheme is used for the Laplacian of the vorticity.

**Data.** Two turbulent flow scenarios are considered with Reynolds numbers of $Re = 16000$ and $Re = 32000$. We generate three independent pairs of LR and HR simulations with spatial grids of $512^2$ and $2048^2$, respectively. LR and HR simulations in each pair start from the same initial conditions. We choose the paired LR and HR snapshots with their vorticity correlation no less than 0.75. The LR inputs and the HR counterparts consist of three channels, each representing a distinct physical quantity. Specifically, these channels denote two velocity variables in the $x$ and $y$ directions, as well as with the vorticity field.

In our `SuperBench`, we use full-field fluid simulations with dimensions of $2048 \times 2048$. For evaluating bicubic and uniform down-sampling degradation methods, we randomly select 1000 and 200 snapshots from the first trajectory for training and validation, respectively, and 200 snapshots from the second trajectory for testing. Additionally, we consider a practical degradation scheme using LR simulation data as inputs. For this, we randomly sample 1200

LR-HR pairs of snapshots from the first two trajectories for training and validation, and 200 snapshots from the third simulation for testing.

### 3.1.2 Cosmology Hydrodynamics

The large-scale structure of the universe is shaped by the parameters of a given cosmological model, as well as by initial conditions. By comparing observational maps of some traces of matter like galaxies or the Lyman $\alpha$ forest (see, e.g., (LSST Science Collaboration, 2009; Aghamousa et al., 2016)) against high-fidelity simulated model universes (see, e.g., (Lukić et al., 2015; Maksimova et al., 2021)), we can constrain the parameters of our cosmological model, such as the nature of dark matter and dark energy, the history of inflation, the reionization in the early universe, or the mass of neutrino particles (Alvarez et al., 2022). In this work, we will use simulation data from Nyx, a massively parallel multiphysics code, developed for simulations of the Lyman $\alpha$ forest. The Nyx code (Almgren et al., 2013) follows the evolution of dark matter modeled as self-gravitating Lagrangian particles, while baryons are modeled as an ideal gas on a set of rectangular Cartesian grids. Besides solving for gravity and the Euler equations, the code also includes the physical processes relevant for the accurate representation of the Ly$\alpha$ forest: chemistry of the gas in the primordial composition of hydrogen and helium, inverse Compton cooling off the microwave background, while keeping track of the net loss of thermal energy resulting from atomic collisional processes (Lukić et al., 2015). All cells are assumed to be optically thin to ionizing radiation, and radiative feedback is accounted for via a spatially uniform, time-varying ultra-violet background radiation. The intricate interactions among diverse physical processes in cosmology data give rise to highly complex multi-scale features, which pose challenges for SR methods.

**Data.** We generate two independent pairs of LR and HR simulations with $512^3$ and $4096^3$ resolution elements (Jacobus et al., 2023), respectively. LR and HR simulations in each pair start from identical initial conditions, and the pairs differ in the random realization but share the same physical and cosmological parameters. Both datasets comprise temperature and baryon density variables. Note that SuperBench currently focuses on 2D SR tasks for cosmology data since the majority of SR research in SciML has concentrated on 2D scenarios. This is due to the reduced computational complexity and the relative ease of training for 2D models, making them an accessible starting point for advancing SR techniques. The goal of SuperBench is to provide a standardized platform for comparing widely used SR methods and will serve as a foundation for extending these approaches to more computationally intensive 3D tasks.

The 2D HR slices are obtained by extracting a fixed sub-domain from the original simulations with a spatial resolution of $2048 \times 2048$ along the $x$-axis. Note that the training/validation and testing datasets are from two independent pairs of simulations. The first dataset aims to test the degradation methods of uniform/bicubic down-sampling. Additionally, we offer a cosmology dataset that uses LR simulation data as inputs to evaluate spatial SR methods in practical applications. The LR simulation is generated with a $512^3$ grid, and the LR counterparts in SuperBench are selected from the corresponding LR sub-region, with a spatial resolution of $256^2$. This dataset serves as a specific data degradation method in the field of SR, which imitates real-world simulation scenarios. It is noteworthy that the temperature and

baryon density in `SuperBench` are presented in the logarithmic space due to their significant magnitudes.

### 3.1.3 WEATHER

Global weather spatiotemporal patterns exhibit highly complex interactions between several physical processes that include turbulence, multi-scale fluid flows, radiation/heat transfer, and multi-phase chemical and biological physics across the atmosphere, ocean, and land surfaces. These interactions span a wide range of spatial and temporal scales that extend over $\mathcal{O}(10)$ orders of magnitude. For instance, spatial scales can span from micrometers (highly localized fluid physics) to thousands of kilometers (full planetary scales). In this work, we consider ERA5 (Hersbach et al., 2020) (downloaded from the Copernicus Climate Change Service (C3S) Climate Data Store), a publicly available dataset from the European Centre for Medium-Range Weather Forecasts (ECMWF). It comprises hourly estimations of multiple atmospheric variables and covers the region from the Earth's surface up to an altitude of approximately 100 km (discretized at 37 vertical levels) with a spatial resolution of 0.25° (25 km) (in latitude and longitude). When represented on a cartesian grid, these variables are a 720 × 1440 pixel field at any given altitude. ERA5 encompasses a substantial temporal scope, spanning from the year 1979 to the present day, and it is generated by assimilating observations from diverse measurement sources with a SOTA numerical model (solver) using a Bayesian estimation process (Kalnay, 2003). It represents an optimal reconstruction of the observed history of the earth's atmospheric state.

**Data.** We provide one dataset that is a subset of ERA5 and consists of three channels for conducting the SR experiments: Kinetic Energy (KE) at 10m from the surface, defined as $\sqrt{u^2 + v^2}$, where $u$ and $v$ are the wind velocity components at 10m altitude; the temperature at 2m from surface; and total column water vapor. These three quantities represent different and crucial prognostic variables—wind velocities are critical for wind and energy resource planning and typically need high resolutions to forecast accurately; surface temperatures are widely tracked during extreme events such as heat waves and for climate change signals; and total column water vapor is diagnostic of precipitation that usually manifests small scale features. The variables are sampled at a frequency of 24 hours (at 00:00 UTC everyday) for 7 years.

### 3.2 Data Preprocessing

It should be noted that the range of the scientific data provided by `SuperBench` is not limited to the common interval of $[0, 255]$, which is typically associated with image data in computer vision. Instead, within the `SuperBench` dataset, we encounter a broad range of data magnitudes, and this presents challenges when assessing baseline performance. To overcome this challenge and ensure a fair evaluation, we standardize the input and target data, using the mean and standard deviation specific to the dataset being evaluated. By normalizing the data, we bring it within a consistent range, allowing for a standardized assessment of baseline performance. Therefore, the evaluation results accurately reflect the relative performance of the baseline methods on the normalized data, ensuring a reliable and robust assessment. In addition, detailed information regarding the preprocessing of

datasets and the creation of interpolation and extrapolation validation/test sets can be found in Appendix A.

## 4 Evaluation Metrics

In our study, we evaluate a range of SOTA SR methods (see below) to establish a solid baseline for the presented problems. To assess the performance of these methods, we employ three distinct types of metrics: pixel-level difference metrics; human-level perception metrics; and domain-motivated error metrics. The detailed metric guide is provided in Table 2.

The pixel-level difference metrics enable us to evaluate quantitatively the SR algorithms by measuring the disparities between the HR ground truth and the generated super-resolved images. These metrics, such as mean squared error (MSE) and peak signal-to-noise ratio (PSNR), provide objective assessments of the fidelity and accuracy of the reconstructed details. By leveraging these metrics, we gain valuable insights into the overall reconstruction quality of the SR methods. The human-level perception metrics permit us to go beyond solely relying on pixel-level metrics that may not capture the perceptual quality of the super-resolved images, as human perception often differs from pixel-wise differences. In particular, by considering human perception, we can evaluate the SR methods based on their ability to produce visually pleasing results that align with human expectations. Finally, we recognize the importance of domain-specific evaluations in scientific applications. Hence, we use domain-motivated error metrics that are tailored to the specific requirements and constraints of the scientific domains under consideration. Incorporating such domain-specific metrics allows us to assess the suitability and effectiveness of SR methods for scientific research purposes.

While our evaluation framework includes standardized metrics that provide a holistic understanding of SR method performance for scientific applications, clearly researchers may have unique research questions and requirements that call for the use of custom metrics. To facilitate such scenarios, our `SuperBench` framework offers a user-friendly interface for defining and incorporating custom evaluation metrics. This flexibility empowers researchers to tailor the evaluation process to their specific needs and explore novel metrics that address the nuances of their research questions.

**Pixel-level difference.** To assess the pixel-level differences between the predicted HR data $\hat{y}$ and the ground truth data $y$, we employ two key metrics: the relative Frobenius norm error (RFNE) and the PSNR. These metrics are defined as

$$\text{RFNE}(y, \hat{y}) = \frac{\|y - \hat{y}\|_{\text{F}}}{\|y\|_{\text{F}}}, \quad \text{and} \quad \text{PSNR}(y, \hat{y}) = 10 \cdot \log_{10} \frac{\max(y)^2}{\text{MSE}(y, \hat{y})}, \tag{2}$$

where $\| \cdot \|_{\text{F}}$ denotes the Frobenius norm, $\text{MSE}(\cdot)$ is the mean-squared error, and $\max(\cdot)$ denotes the maximum value. By quantifying the differences between the predicted and ground truth data, these metrics enable a comprehensive assessment of the pixel-level fidelity achieved by the SR algorithms. They provide valuable information for evaluating the accuracy and effectiveness of SR methods, and they are relevant for applications that demand high

Table 2: Summary of metric guide in `SuperBench`

| Datasets | Pixel-level difference | Human-level perception | Domain-motivated error metrics |
|---|---|---|---|
| Fluid Flow data | MAE, MSE, RFNE, IN, PSNR | SSIM | Continuity, Energy spectrum |
| Cosmology data | MAE, MSE, RFNE, IN, PSNR | SSIM | - |
| Weather data | MAE, MSE, RFNE, IN, PSNR | SSIM | ACC |

precision in pixel-wise reconstruction. In addition, we also consider the infinity norm (IN) as a metric to assess extreme statistical properties.

**Human-level perception.** The impact of minor perturbations and content shifts on the signal $y$ can lead to substantial degradation in both RFNE and PSNR, even when the underlying content or patterns remain unchanged (Agustsson and Timofte, 2017). Hence, there is a growing interest in evaluating SR algorithms in a structural manner that aligns with human perception. To this end, the structural similarity index measure (SSIM) (Wang et al., 2004, 2020c) can be used as a metric for evaluating SR algorithms. SSIM is a perception-based metric that focuses on image and graphical applications. In contrast to metrics like RFNE and PSNR, which primarily measure the pixel-wise discrepancies between the super-resolved data and their ground truth counterparts, SSIM takes into account the structural information and relationships within the images. By considering the structural characteristics of the data, SSIM provides a more nuanced evaluation of the SR algorithms, capturing perceptual similarities that go beyond pixel-level differences.

**Domain-motivated error metrics.** Within our `SuperBench` framework, we provide researchers with the flexibility to incorporate domain-motivated error metrics. This is particularly valuable in scientific domains where prior knowledge is available. These metrics allow for a more comprehensive evaluation of SR methods by considering domain-specific constraints and requirements. In scientific domains such as fluid dynamics, where the preservation of continuity and conservation laws is crucial, it becomes essential to assess the SR algorithms from such a physical perspective (Wang et al., 2020b; Esmaeilzadeh et al., 2020). Researchers can incorporate evaluation metrics that focus on physical aspects and examine the reconstructed variables to ensure they adhere to these fundamental principles. In this paper, we introduce a physics error metric to measure the preservation of continuity property in fluid flow datasets (Wang et al., 2020b), and we also visualize the energy spectrum for comparative analysis. Similarly, in climate science, the evaluation metrics often involve multi-scale analysis due to the presence of multi-scale phenomena that are ubiquitous in this field. Researchers may employ metrics such as the Anomaly Correlation Coefficient (ACC) (Rasp et al., 2020) to evaluate the performance of SR methods. These metrics enable the assessment of how well the SR algorithms capture and represent the complex, multi-scale features present in climate data.

Table 3: Overview of our assessment on baseline models for each dataset (✓✓: Excellent, ✓: Good, ◯: Acceptable, ✗: Suboptimal, ✗✗: Bad).

| Aspect | Interp. | SRCNN | Sub-pixel CNN | SRGAN | EDSR | WDSR | FNO | SwinIR |
|---|---|---|---|---|---|---|---|---|
| Fluid flow | ✗✗ | ✗ | ◯ | ✗ | ✓ | ✓ | ◯ | ✓✓ |
| Cosmo. | ✗✗ | ✗ | ◯ | ◯ | ◯ | ◯ | ◯ | ✓ |
| Weather | ✗✗ | ✗ | ◯ | ◯ | ◯ | ◯ | ◯ | ✓ |
| Upsampling (×8) | ✗✗ | ✗ | ◯ | ◯ | ✓ | ◯ | ◯ | ✓✓ |
| Upsampling (×16) | ✗✗ | ✗✗ | ◯ | ◯ | ◯ | ◯ | ◯ | ◯ |
| LR Sim. | ✗✗ | ✗ | ◯ | ◯ | ◯ | ◯ | ◯ | ◯ |
| Physics Preservation | ✗ | ✗✗ | ◯ | ✗✗ | ◯ | ✗ | ✗✗ | ✓ |

## 5 Experiments and Analysis

This section presents our experimental setup and performance analysis of baseline models using the `SuperBench` datasets. The aim of `SuperBench` is to provide more challenging and realistic SR settings, considering the remarkable progress achieved in SISR research (Moser et al., 2023). To accomplish this goal, we incorporate various data degradation schemes within the `SuperBench` framework. These schemes simulate realistic degradation scenarios encountered in scientific applications. Specifically, we consider the following baseline models: Bicubic interpolation; SRCNN (Dong et al., 2015); Sub-pixel CNN (Shi et al., 2016); SRGAN (Ledig et al., 2017); EDSR (Lim et al., 2017); WDSR (Yu et al., 2018); Fourier Neural Operator (FNO) (Li et al., 2021); and SwinIR (Liang et al., 2021). See Appendix B and C for detailed information regarding the baseline models and the corresponding training protocol. An overview of baseline performance on different aspects is provided in Table 3.

### 5.1 Evaluation Setup

In `SuperBench`, we define three distinct data degradation scenarios for spatial SR tasks, each designed to model a specific scientific situation: (i) The general computer vision scenario, which involves bicubic down-sampling. This scenario serves as a standard degradation method for SR evaluation in various image processing applications. (ii) The uniform down-sampling scenario, which considers noise in addition to down-sampling, mimicking the experimental measurement process in scientific domains. This scenario aims to replicate the challenges of accurately reconstructing data from low-fidelity measurements obtained by experimental sensors. (iii) The direct use of LR simulation data as inputs, which is specific to scientific modeling. This scenario explores the performance of SR algorithms when provided with LR data generated through simulation processes.

**Up-sampling factors.** For scenarios (i) and (ii), `SuperBench` offers two tracks of up-sampling factors: ×8 and ×16. These factors determine the levels of up-sampling required to recover the HR details from the degraded LR inputs. We consider a scaling factor of ×16, motivated by the growing interest in significant up-sampling factors for scientific SR (Gao et al., 2021). For scenario (iii), we specifically test the ×8 up-sampling on the fluid and cosmology datasets using LR simulation data with a spatial resolution of $256 \times 256$ as inputs.

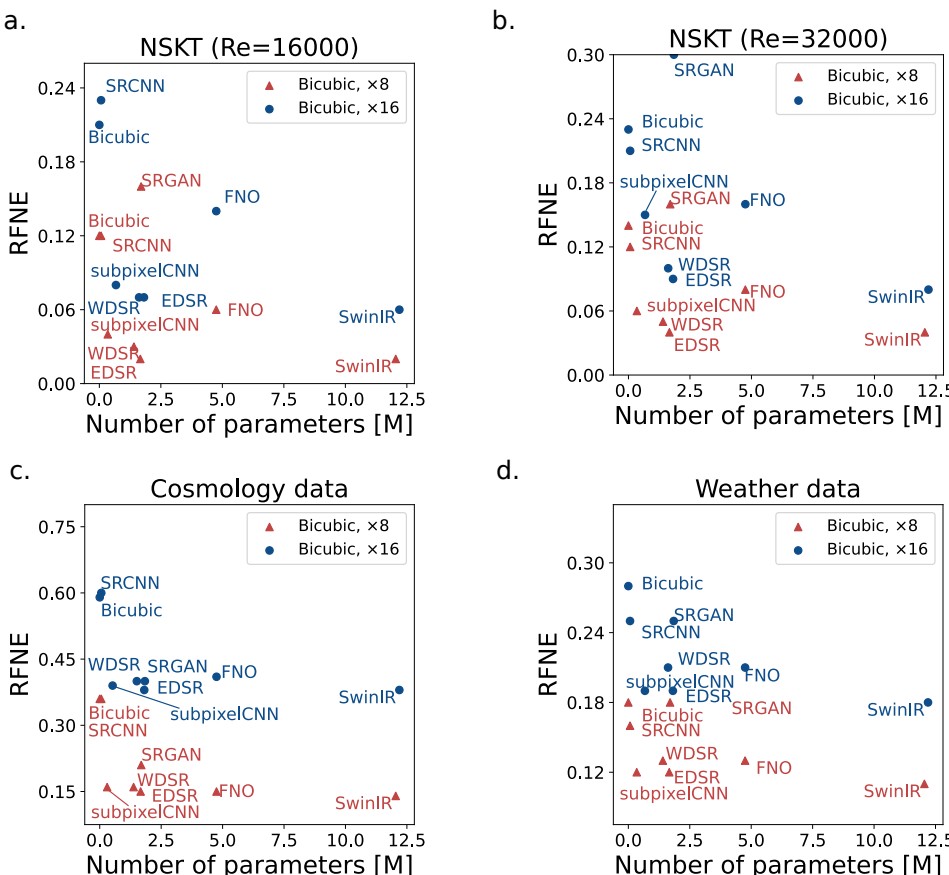

Figure 4: The results of RFNE versus model parameters on four datasets considering scenario (i) with up-sampling factors ×8 and ×16.

The HR counterparts in this scenario have an exceptionally high resolution of $2048 \times 2048$, representing the challenges associated with SR tasks in the cosmology domain.

**Data noise.** Recognizing the presence of noise in scientific problems, we provide the option to evaluate the performance of the `SuperBench` dataset under noisy LR scenarios, specifically in scenario (ii). This feature aligns with the scientific requirement for accurate data reconstruction from low-fidelity measurements obtained by experimental sensors, ensuring a faithful representation of the underlying physical phenomena. The noise level in the dataset is defined by the channel-wise standard deviation of the specific dataset. Additionally, users have the flexibility to define custom noise ratios of interest. In our `SuperBench` experiments, we test cases with noise levels set at 5% and 10% to assess the robustness and performance of SR methods under different noise conditions.

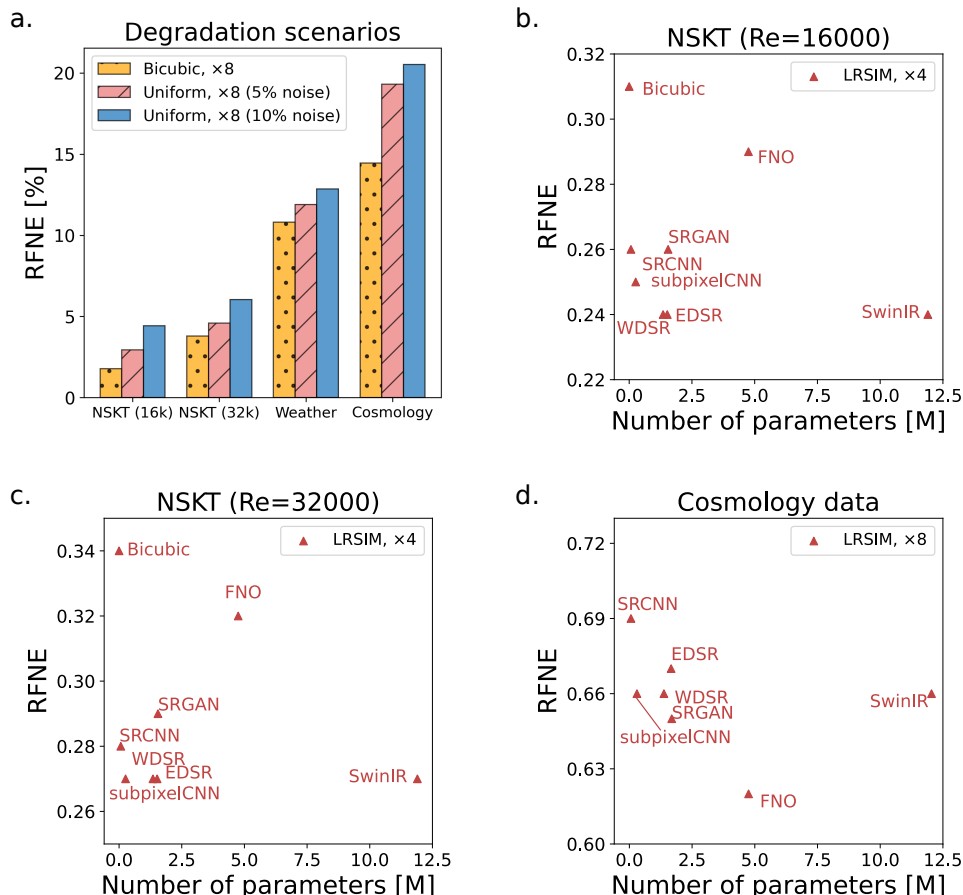

Figure 5: Comparative results of different degradation methods. (a) exhibits the RFNE results from scenarios (i) and (ii) using the SwinIR model, with up-sampling factors ×8. (b-d) show the results of scenario (iii) with LR simulation data as inputs.

## 5.2 Results

**General performance.** Figure 4 presents the performance of baseline models on four datasets considering degradation scenario (i) with up-sampling factors ×8 and ×16. Overall, the baseline SR methods achieve good pixel-level accuracy for super-resolving fluid flow datasets, but they fail to perform well on both cosmology and weather data. This discrepancy arises from more complex multi-scale structures and variations inherent in cosmology and weather data. As shown in Figure 4(c), the cosmology data is the most challenging by measuring the RFNE values among all datasets. SwinIR exhibits performance comparable to residual networks (EDSR and WDSR) on two NSKT datasets, while achieving SOTA results on cosmology and weather data. This superior performance can be attributed to its specialized network design, which effectively captures multi-scale features (Liang et al., 2021). Additionally, we observe that SRCNN and SRGAN demonstrate limited robustness across different scientific datasets.

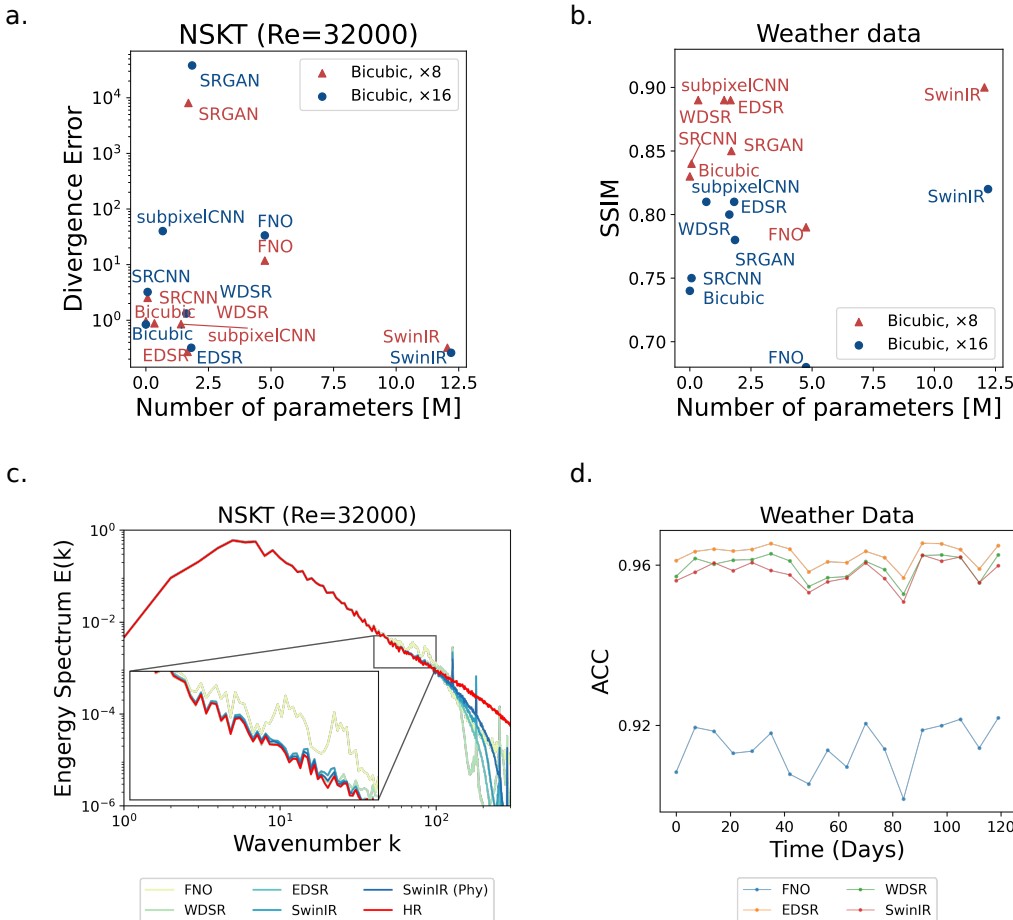

Figure 6: A summary of the evaluations of physical properties and multi-scale features. (a) and (c) show the results of testing physics loss and energy spectrum on the fluid dataset, respectively. SwinIR (Phy) denotes the physics-constrained SwinIR model with continuity loss. (b) and (d) present the results of multi-scale features on the weather dataset, including SSIM and ACC metrics.

**Up-sampling factors.** Figure 4(a-d) illustrates significant RFNE discrepancies between the two up-sampling factors (×8 and ×16) across four datasets. Notably, the ×16 track is substantially more challenging, compared to the ×8 track. To establish a meaningful benchmark, we propose to use ×16 for fluid flow datasets and ×8 for cosmology and weather data. Additionally, we consider ×16 up-sampling for cosmology and weather data as an extreme SR validation.

**Degradation analysis.** We showcase the model performance on different degradation schemes in Figure 5. By using SwinIR tested on the ×8 track as an illustrative case, we present the baseline performance across various degradation scenarios in Figure 5(a). It is clear that scenario (ii), which considers both uniform down-sampling and noise, poses greater challenges, compared to scenario (i) of bicubic down-sampling. The level of difficulty in SR

progressively intensifies with increasing noise. Moreover, Figure 5(b-d) present the baseline performance on fluid and cosmology datasets considering scenario (iii), which directly uses LR simulation data as inputs. The LR simulation data lack the presence of fine-scale features and high-frequency information due to numerical errors, which poses a challenge for SR. All baseline models show inferior performance compared with those from scenarios (i) and (ii). Note that the FNO model demonstrates unstable performance when using LR simulation data as inputs. While it achieves excellent results on cosmology data, its performance on NSKT datasets is unsatisfactory. This instability is attributed to the selection of Fourier modes in FNO, making it struggle to capture high-frequency information effectively. Scenarios (ii) and (iii) are specifically designed to emulate the experimental and simulation conditions, respectively. Despite the inherent difficulties, it is crucial to explore novel SR methods to advance their applicability to real-world scenarios.

**Physics preservation.** We evaluate the baseline performance of the physics error and energy spectrum on fluid flow data. A metric that measures the continuity property in fluid flows is considered, as shown in Figure 6(a). SwinIR performs the best among all baseline models on physics errors, corresponding to the results from pixel-level accuracy. We observe that SRGAN exhibits higher physics errors during evaluation. This is probably due to the inherent randomness of generative models, which can introduce non-physical high-frequency artifacts. Similar findings regarding relatively large physics errors in generative models have been noted in a recent study (Shu et al., 2023). Additionally, achieving satisfactory pixel-level accuracy does not ensure the preservation of underlying physical properties. The physics loss of several deep learning-based models, such as FNO and WDSR ($\times 16$), is worse than that of the standard Bicubic interpolation, although they achieve better RFNE accuracies. Therefore, although previous SR methods have demonstrated notable achievements in terms of pixel-level difference and human-level perception, there is a need for scientific SR methods that respect the underlying physical laws of problems of interest. This is particularly important in light of recent results highlighting methodological challenges in delivering on the promise of SciML (Krishnapriyan et al., 2021; Edwards, 2022; Krishnapriyan et al., 2022).

We implement a physics-constrained SwinIR model with a physics regularizer of the continuity loss. The corresponding weighting coefficient $\lambda_p$ is set as 0.001. We show a comparative study of the physics-constrained SwinIR model and other representative baseline models in terms of the energy spectrum. Figure 6(c) demonstrates a better alignment of physics-constrained SwinIR with the ground truth compared with that of the SwinIR model, which outperforms the rest of the baseline models. It validates the effectiveness of incorporating physics loss into deep learning models with a better alignment with the ground truth.

**Multi-scale details.** Comparative snapshots of baseline models against the ground truth HR weather data are depicted in Figure 2. The zoomed-in figures demonstrate the limitation of the current SR methods for recovering fine-scale details. In addition, we present the SSIM results for the weather data in Figure 6(b), where the SwinIR holds the best performance (0.90) for the $\times 8$ track in scenario (i). Moreover, Figure 6(d) shows the ACC performance of baseline models along the time, where SwinIR performs the best. However, there remains

ample space for improvement in learning multi-scale features, indicating potential avenues for further advancements in SR algorithms.

## 6 Conclusion

In this paper, we introduce `SuperBench`, a new large-scale and high-quality scientific dataset for SR. We analyze the baseline performance on the `SuperBench` dataset and identify the challenges posed by large magnification factors and the preservation of physical properties in the SR process. Moreover, different data degradation scenarios have been investigated to measure the robustness of the baseline models. The `SuperBench` dataset and our analysis pave the way for future research at the intersection of computer vision and SciML. We anticipate that our work will inspire new methodologies (e.g., constrained ML) to tackle the unique requirement of SR in scientific applications.

`SuperBench` provides a flexible framework and provides the option to be extended to include scientific data from other domains (e.g., solid mechanics). This expansion can help to facilitate the assessment of SR algorithms in diverse scientific contexts and foster the development of tailored solutions. In addition, `SuperBench` can be extended from spatial SR tasks to temporal and spatiotemporal SR tasks, and additional data degradation methods can be considered to further emulate the real-world challenges. Lastly, we plan to incorporate HR 3D datasets, such as JHTDB (Li et al., 2008) and BLASTNet (Chung et al., 2024), into `SuperBench` since real-world scientific applications often handle 3D data. Meanwhile, we will adapt the baseline models to be compatible with 3D SR tasks.

Moreover, it is important to discuss potential negative uses and pitfalls. For example, SR methods could introduce artifacts or inaccuracies that may mislead scientific conclusions if used without sufficient validation. This is particularly important in some critical scientific and engineering domains, such as aerospace, earthquake, and weather modeling, where erroneous results could have significant real-world consequences. To ensure responsible use of the benchmark, we emphasize the importance of rigorous testing and validation to mitigate these risks before deploying any models in real-world applications. We also acknowledge the potential for misuse if models trained on our datasets are applied without considering their limitations or if results are interpreted without the appropriate domain expertise.

## Acknowledgments and Disclosure of Funding

We would like to sincerely acknowledge the constructive discussion with Dr. Dmitriy Morozov. This work was supported by the U.S. Department of Energy, Office of Science, Office of Advanced Scientific Computing Research, Scientific Discovery through Advanced Computing (SciDAC) program, under Contract Number DE-AC02-05CH11231, and the National Energy Research Scientific Computing Center (NERSC), operated under Contract No.DE-AC02-05CH11231 at Lawrence Berkeley National Laboratory.

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

## Appendix A. Additional Data Details

In `SuperBench`, we provide two perspectives to evaluate the model performance: interpolation and extrapolation. The goal of extrapolation datasets is to measure the generalizability of baseline models to future or domain-shifted snapshots, which is similar to the testing processing in general computer vision (CV) tasks. Interpolation datasets aim to assess the model performance on intermediate snapshots, either in terms of time or space. Notably, the baseline performances shown in Experiments and Analysis (Section 5) are all extrapolation results. The specific information regarding the design of interpolation and extrapolation data for each dataset is presented below.

**Fluid flow data.** The validation and testing datasets used for evaluation are sourced from the same domain as the training dataset, specifically the $[0, 2\pi]^2$ region with a spatial resolution of $2048 \times 2048$. Within this context, interpolation refers to data that are sampled from the same simulation as the training data. Extrapolation, on the other hand, involves data obtained from simulations that are generated with different initial conditions.

**Cosmology data.** There are two independent pairs of low-resolution (LR) and high-resolution (HR) simulations with $512^3$ and $4096^3$ resolution elements, respectively. The training dataset and interpolation datasets are from the same data pair, and the extrapolation datasets are defined from the other data pair.

**Weather data.** The training datasets are selected from the years 2008, 2010, 2011, and 2013. The validation datasets for interpolation and extrapolation evaluation focus on the years 2012 and 2007 (look-back test), respectively. For the testing datasets, we employ the data from the years 2009 and 2014, 2015 for the corresponding interpolation and extrapolation tasks.

## Appendix B. Baseline Models

We consider the following state-of-the-art (SOTA) super-resolution (SR) methods as baselines.

- **Bicubic interpolation.** Bicubic interpolation is one of the most widely used methods for image SR due to its simplicity and ease of implementation. In addition, the capability of bicubic interpolation to upsample spatial resolution serves as a cornerstone for many deep learning-based SR techniques (Dong et al., 2015; Kim et al., 2016).

- **SRCNN.** SRCNN (Dong et al., 2015) is the first work to apply deep convolutional neural networks (CNNs) for learning to map the patches from low-resolution (LR) to high-resolution (HR) images. It outperforms many traditional methods and spurs further advancements in DL-based SR frameworks.

- **Sub-pixel CNN.** Shi *et al.* (Shi et al., 2016) propose a new method for increasing the resolution using pixel-shuffle. It enables training deep neural networks in LR latent space, and it also achieves satisfactory reconstruction performance, which improves computational efficiency.

- **SRGAN.** Ledig *et al.* (Ledig et al., 2017) introduce a generative adversarial network (GAN) with a perceptual loss function composed of an adversarial loss and a content loss for image SR. SRGAN is a generative model-based SR method that has performed well in reconstructing original HR images.

- **EDSR.** Using a deep residual network architecture that encompasses an extensive number of residual blocks, EDSR (Lim et al., 2017) can efficiently learn the mapping between LR and HR images as well as capture hierarchical features. EDSR has demonstrated remarkable performance in generating high-quality SR images, and it remains a prominent benchmark in this field.

- **WDSR.** Yu *et al.* (Yu et al., 2018) further improves the reconstruction accuracy and computational efficiency by considering wider features before ReLU in residual blocks. WDSR has achieved the best performance in NTIRE 2018 Challenge on single-image super-resolution (SISR) (Timofte et al., 2018).

- **FNO.** FNO (Li et al., 2021) leverages the Fourier transform to obtain the integral kernel operators, enabling efficient learning of mappings in function spaces. This approach significantly reduces computational complexity and achieves state-of-the-art performance in various super-resolution tasks by handling multiscale features effectively.

- **SwinIR.** SwinIR (Liang et al., 2021) is based on advanced Swin Transformer (Liu et al., 2021) architecture. The Swin Transformer layers are used for local attention and cross-window interaction. It largely reduces the number of parameters and also achieves SOTA performance in SISR.

## Appendix C. Training Details

In this section, we provide additional training details for all baseline models. Training the models directly on the high-resolution data is challenging and infeasible due to memory constraints even using an A100 GPU with 40GB. To this end, we randomly crop each high-resolution snapshot into a smaller size for training. For all datasets in `SuperBench`, the patch size is defined as $128 \times 128$. The number of patches per snapshot is selected as 8. We preprocess the data by standardizing each dataset in `SuperBench`, i.e., we subtract the mean value and divide it by the standard deviation. All models are trained from scratch on Nvidia A100 GPUs. The training code and configurations are available in our GitHub repository.

**SRCNN.** We use the default network design in the SRCNN paper (Dong et al., 2015). In addition, we substitute the original Stochastic gradient Descent (SGD) optimizer with the ADAM (Kingma and Ba, 2015) optimizer. The learning rate is set as $1 \times 10^{-3}$ and the weight decay is $1 \times 10^{-5}$. We train the SRCNN models for 200 epochs. The other hyper-parameters and training details follow the original implementation. The batch size is set to 32 and Mean Squared Error (MSE) is employed as the loss function.

**Sub-pixel CNN.** The default network architecture is applied (Shi et al., 2016). The learning rate is set as $1 \times 10^{-3}$ for fluid flow datasets and $1 \times 10^{-4}$ for cosmology and weather

datasets. The batch size is 32, and the weight decay is $1 \times 10^{-4}$. The training is performed for 200 epochs with the Adam optimizer. Moreover, MSE is considered as the loss function.

**SRGAN.** For the Generator part of SRGAN, we employ 16 residual blocks, each with a hidden channel dimension of 64. A 2D convolutional layer is appended after the final `Tanh` layer of the default generator to map the predicted values to the normalized data space. For the Discriminator, we use a CNN with 4 blocks based on the default network setting. We replace the context loss with the standard MSE loss between generated data and ground truth since a pretrained VGG (Simonyan and Zisserman, 2015) feature extractor is not suitable for scientific data. The input and output channels are set to 2 for the cosmology datasets and 3 for other datasets. The learning rate is configured as $2 \times 10^{-4}$ with a weight decay of $1 \times 10^{-6}$. All the datasets are trained for 600 epochs using Adam optimizer, with a batch size of 512 for parallel training across 4 GPUs.

**EDSR.** We follow the default network setting in EDSR (Lim et al., 2017), which uses 16 residual blocks with the hidden channel as 64. The learning rate is set as $1 \times 10^{-4}$ and the weight decay is $1 \times 10^{-5}$. The batch size is defined as 64. We train the fluid flow datasets for 400 epochs and the cosmology and weather datasets for 300 epochs with the ADAM optimizer. We also follow the default training protocol to use L1 loss as the objective function.

**WDSR.** We consider the WDSR-A (Yu et al., 2018) architecture for the SR tasks in `SuperBench`, which considers 18 light-weight residual blocks with wide activation. The hidden channel is defined as 32. The learning rate and the weight decay are set as $1 \times 10^{-4}$ and $1 \times 10^{-5}$, respectively. We train all the WDSR models with 300 epochs by using the ADAM optimizer. The batch size is selected as 32 and L1 loss is also used.

**SwinIR.** We follow the default network hyperparameters and training protocol for classical and real-world image SR in SwinIR (Liang et al., 2021). The residual Swin Transformer block (RSTB), Swin Transformer layer (STL), window size, channel number, and attention head number are set as 6, 6, 8, 180, and 6, respectively. The learning rate and the weight decay are also chosen as $1 \times 10^{-4}$ and $1 \times 10^{-5}$, respectively. The batch size is set to 32. We use the AdamW (Loshchilov and Hutter, 2019) optimizer to train SwinIR models for 200 epochs. L1 loss is employed for training.

**Physics-constrained SwinIR.** We enhance the performance of the SwinIR model by incorporating physics laws as a soft constraint into the optimization process while maintaining other settings consistent, termed as SwinIR(Phy). The loss function is defined as:

$$\mathcal{L} = \mathcal{L}_d + \lambda_p \mathcal{L}_p \tag{3}$$

where $\mathcal{L}_d$ represents the data loss and $\mathcal{L}_p$ denotes the physics loss. The penalty factor $\lambda_p$ is a tunable hyperparameter. We perform a grid search to determine the optimal value of $\lambda_p$ from the set $\{10, 1, 0.1, 0.01, 0.001, 0.0001\}$. MSE loss is employed for physics loss.

For fluid datasets, the incompressibility condition of the flow is enforced by incorporating the residue of the continuity equation as the physics loss, which is given by

$$\frac{\partial u}{\partial x} + \frac{\partial v}{\partial y} = 0. \tag{4}$$

We approximate the spatial derivatives in Eq. 4 using the finite difference method. Specifically, weighted gradient-free convolution kernels are applied to the snapshots to compute these derivatives. The size and values of the kernel depend on the desired order of accuracy of the central difference scheme. In this SwinIR(Phy) model, we employ second-order kernels.

**FNO.** We perform a grid search to obtain the best hyperparameter configurations. We consider the number of Fourier Layers $L = [1, 4, 7]$, the number of hidden variables $N = [20, 40, 64]$, and the maximum cutoff frequency mode $M = [3, 12, 20]$. The best configuration is found to be $[L, N, M] = [4, 64, 12]$. The remaining hyper-parameter setting and network implementation are consistent with the original implementation of FNO (Li et al., 2021). Since FNO models require inputs and outputs with the same dimensions, we upscale the model inputs with Bicubic interpolation. We train the model for 500 epochs using the ADAM optimizer and a learning rate scheduler with a step size of 100, a decay rate of 0.5, and an initial learning rate of $1 \times 10^{-3}$. The MSE loss is employed and the batch size is selected as 32. However, unlike other models, which can be trained on small patches (e.g., target resolution $128^2$) and still perform well when directly tested on much larger resolutions (e.g., target resolution $2048^2$), the performance of FNO models significantly deteriorates under the same testing conditions. To address this issue, we modified the FNO by incorporating a patch-splitting and patch-merging module. Both modules allow the model to evaluate large snapshots by processing them in evenly separated, non-overlapping patches and then merging the predicted patches back into large snapshots for post-analysis.

## Appendix D. Additional Results

In this section, we show additional results to support and complement the findings in Experiments and Analysis (Section 5). We show example snapshots for visualizing and comparing baseline performances under three degradation scenarios. We also conduct a comprehensive analysis across different evaluation metrics.

### D.1 Results Visualization

This section presents visualization results to demonstrate the reconstruction performance of baseline models. Three degradation scenarios are considered. As shown in Figures 7-10, we show the baseline performance by using bicubic down-sampling degradation. For the fluid flow datasets, EDSR and SwinIR can recover the HR representation from LR inputs very well by a factor of $\times 8$, but it is challenging to achieve satisfactory results for the $\times 16$ up-sampling task. For cosmology and weather datasets, all baseline models exhibit limitations in effectively reconstructing the multi-scale features.

Figures 11-14 show the $\times 8$ SR results of SwinIR under the degradation scenario of combining uniform down-sampling and noise (5% and 10%). SwinIR can capture the fine-scale features

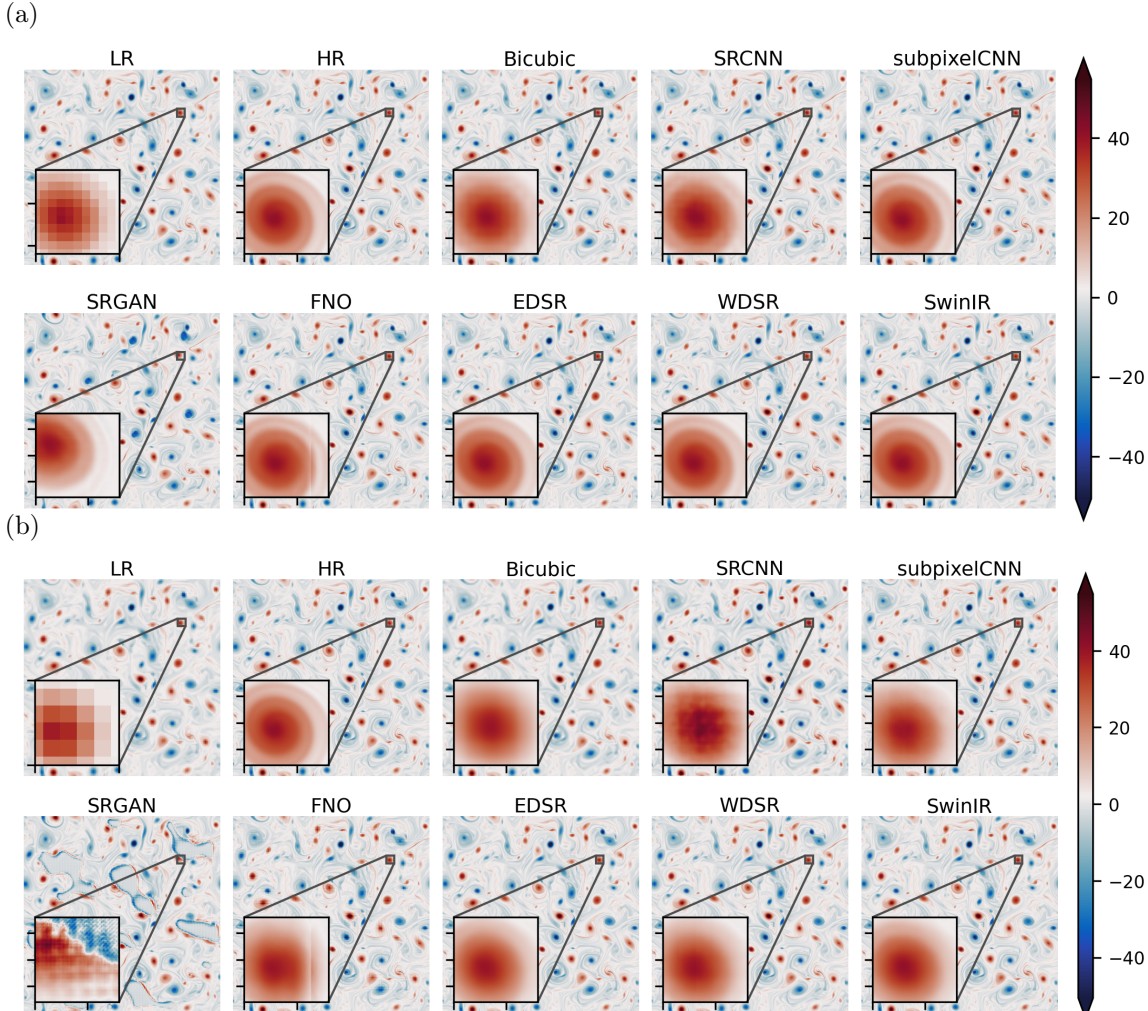

Figure 7: Showcasing baseline SR methods on turbulent fluid flow data ($Re = 16000$) under bicubic down-sampling. Here (a) and (b) represent the results of ×8 and ×16 up-sampling tasks.

of the fluid flow datasets, but it shows limited reconstruction capability for cosmology and weather data.

Figure 15-17 showcases the baseline results on the fluid and cosmology data, using LR simulation data as inputs. The LR input data, characterized by a lack of fine-scale and high-frequency information, poses challenges in recovering the corresponding HR counterparts. As a result, all baseline models present limitations in this SR track.

## D.2 Detailed Baseline Performance

The baseline results of all experiments are summarized in the following tables. The error metrics include RFNE (↓%), infinity norm (IN↓), PSNR (↑dB), and SSIM (↑). In addition, we also evaluate the performance of physics preservation on fluid flow datasets by measuring

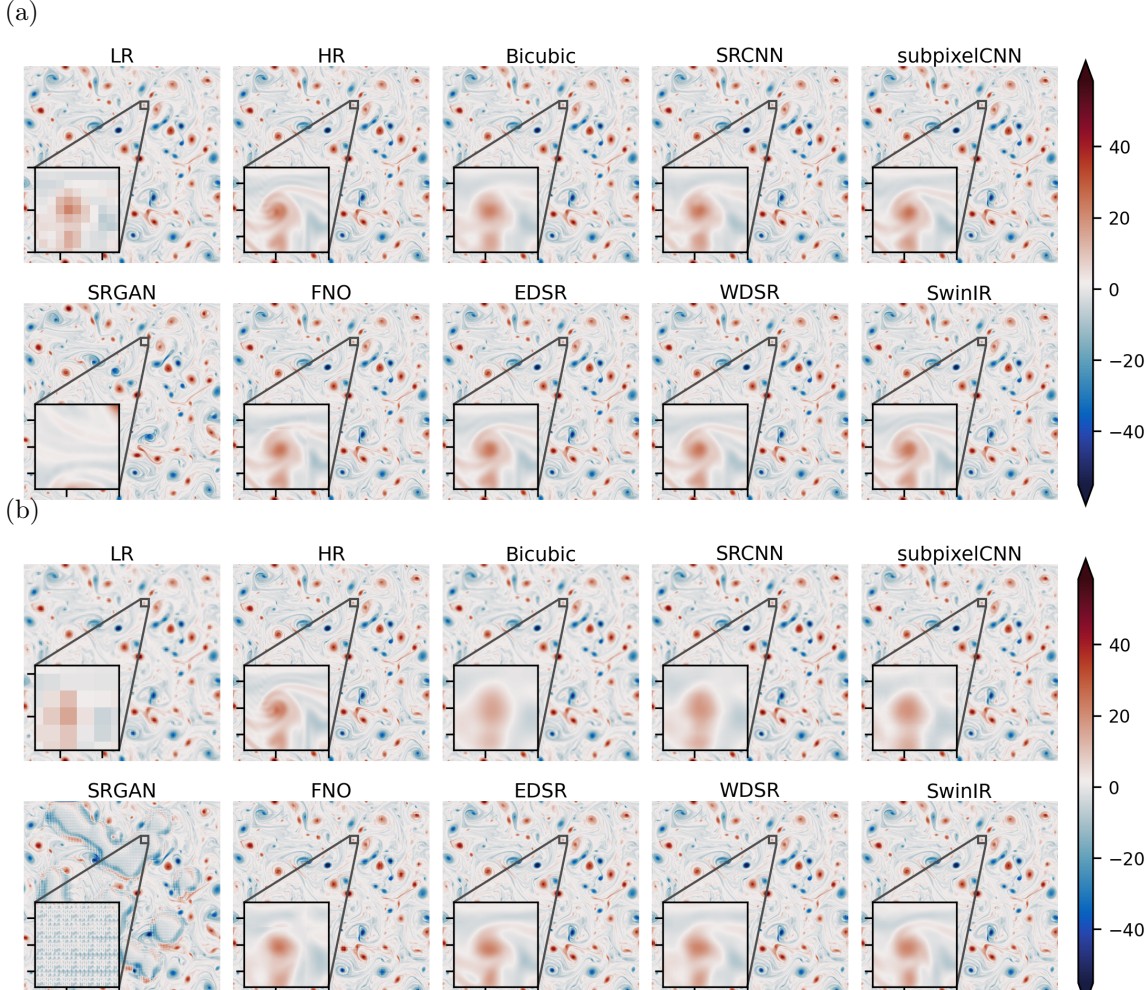

Figure 8: Showcasing baseline SR methods on very turbulent fluid flow data ($Re = 32000$) under bicubic down-sampling. Here (a) and (b) represent the results of ×8 and ×16 up-sampling tasks.

continuity loss, as shown in Table 13. The baseline models demonstrate superior SR performance in terms of interpolation errors, indicating the relative ease of the interpolation task compared to extrapolation.

Tables 4-7 reveal that SwinIR outperforms other baseline models in terms of RFNE and SSIM metrics, except for the ×8 up-sampling task on fluid flow data ($Re = 16000$) and ×16 up-sampling on cosmology data. In terms of PSNR, EDSR exhibits better performance on fluid flow datasets, while SwinIR demonstrates superiority on cosmology (×8) and weather data. These results empirically validate the effectiveness of SwinIR in learning multi-scale features.

Furthermore, Tables 8-9 present the ×8 SR results of SwinIR on the `SuperBench` dataset under noisy scenarios with uniform down-sampling degradation. These scenarios simulate

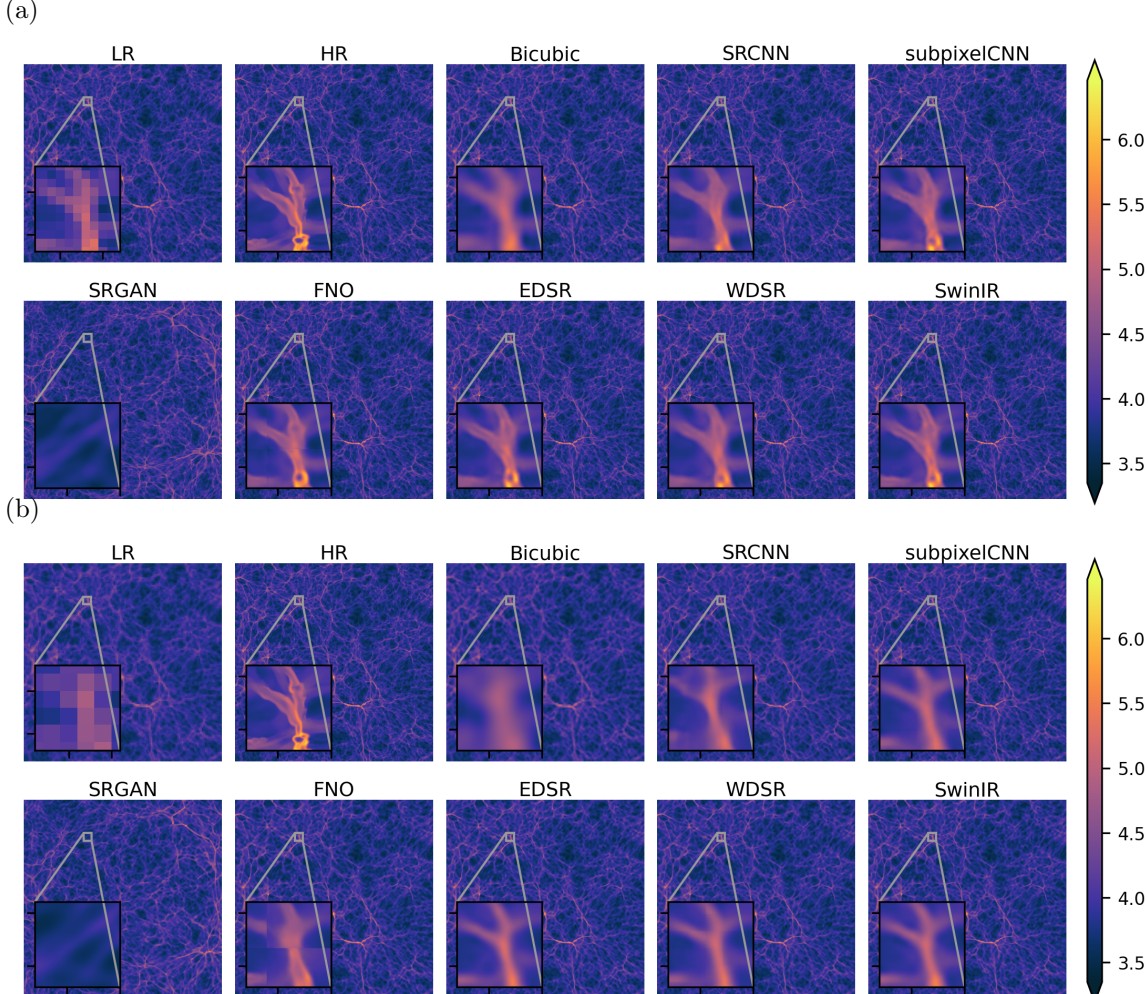

Figure 9: Showcasing baseline SR methods on cosmology data under bicubic down-sampling. Here (a) and (b) represent the results of ×8 and ×16 up-sampling tasks.

real-world experimental conditions, where the existence of noise introduces difficulties in reconstructing high-fidelity scientific data. As the noise level increases, the SR task becomes progressively demanding.

We show the baseline performance of ×8 SR on the LR simulation scenarios in Table 10-12. This task is much more complicated compared to the scenarios involving down-sampling methods for degradation. LR simulation data, in comparison to common down-sampling methods, exhibits a more significant loss of fine-scale and high-frequency features. Moreover, the numerical errors in LR simulations play a crucial role in challenging deep learning models when recovering HR counterparts.

(a)

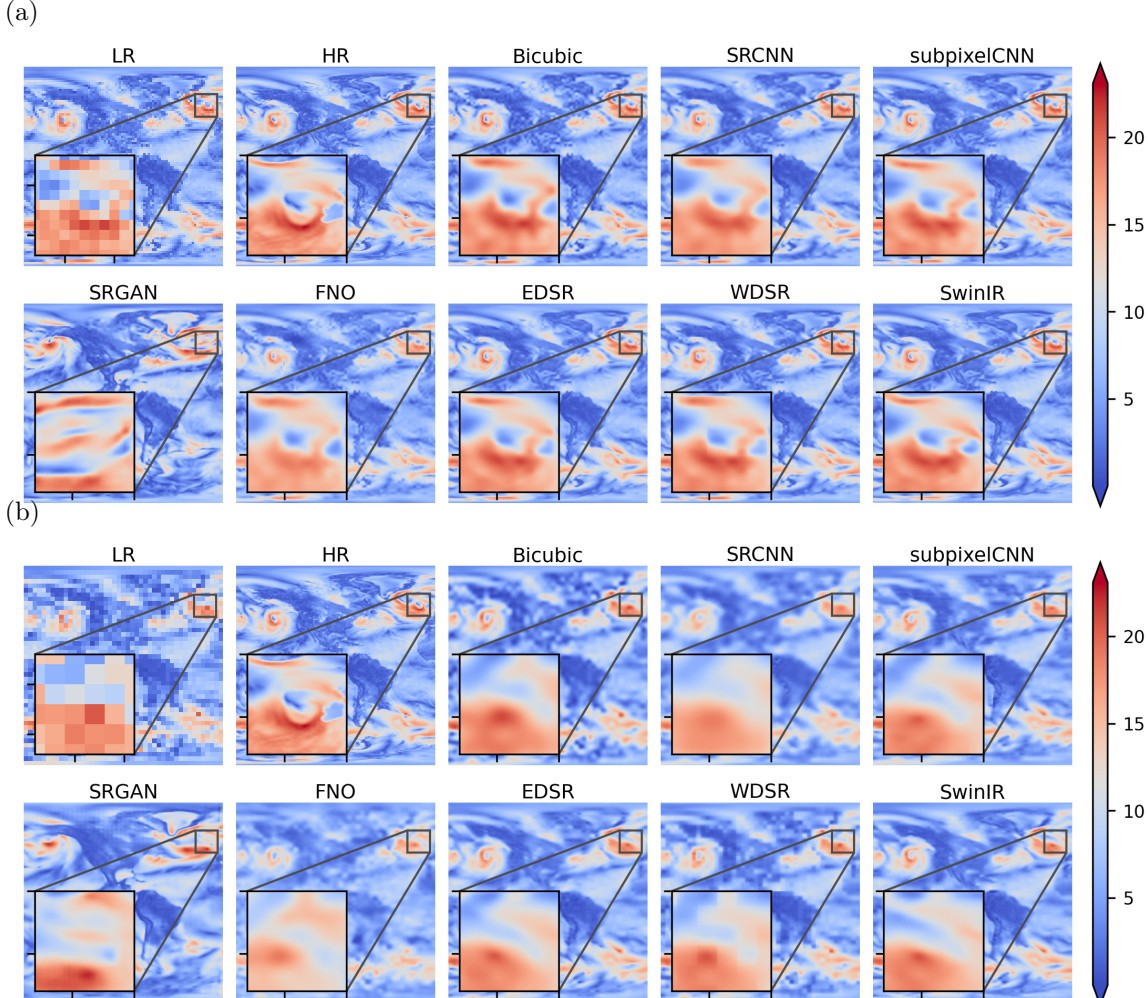

(b)

Figure 10: Showcasing baseline SR methods on weather data under bicubic down-sampling. Here (a) and (b) represent the results of ×8 and ×16 up-sampling tasks.

## Appendix E. Reproducibility

The code for processing the datasets, running baseline models, and evaluating model performance is publicly available in our GitHub repository. The README file contains system requirements, installation instructions, and running examples. Detailed training information is provided in Appendix C.

## Appendix F. Data Hosting, Licensing, Format, and Maintenance

SuperBench is a collaborative effort involving a diverse team from different institutes, including Lawrence Berkeley National Lab (LBNL), University of California at Berkeley, International Computer Science Institute (ICSI), and University of Tennessee, Knoxville. In

(a)

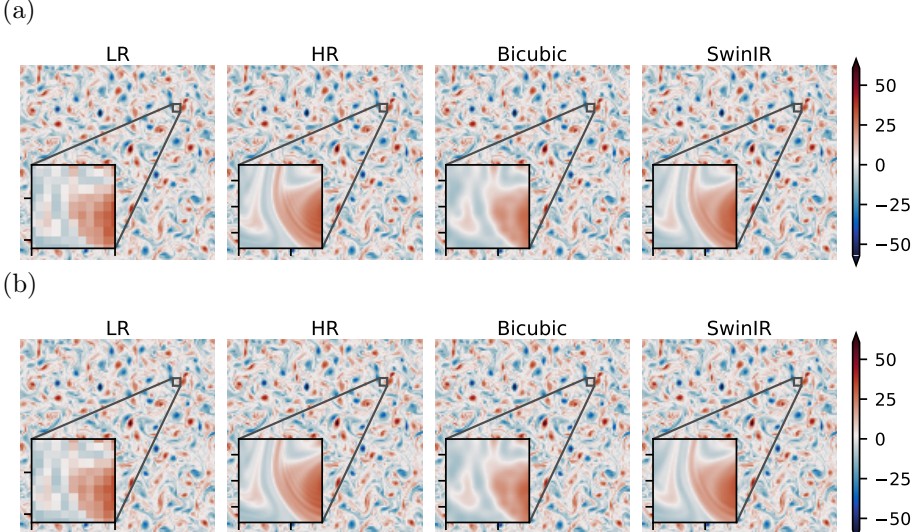

(b)

Figure 11: Showcasing baseline SR methods on fluid flow data ($Re = 16000$) under uniform down-sampling and noise. Here (a) and (b) show the results of ×8 up-sampling considering 5% and 10% noise, respectively.

(a)

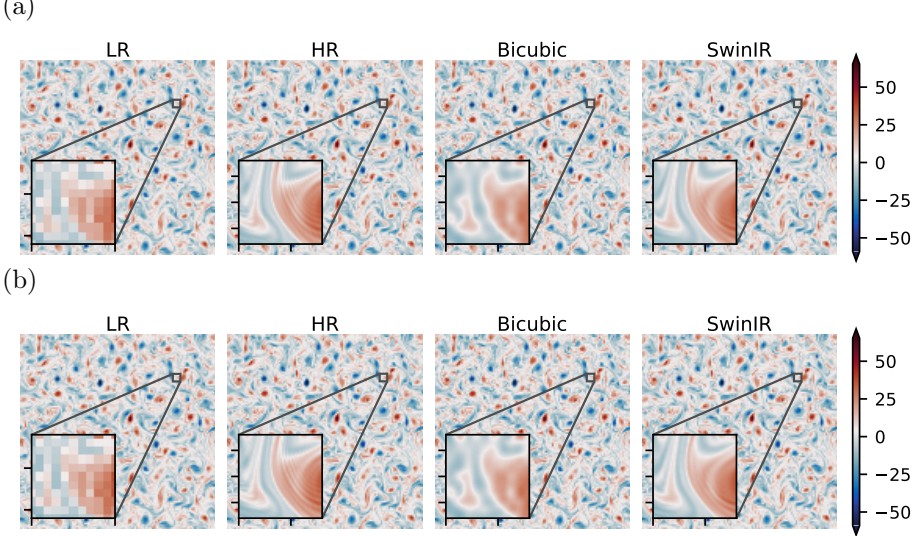

(b)

Figure 12: Showcasing baseline SR methods on fluid flow data ($Re = 32000$) under uniform down-sampling and noise. Here (a) and (b) show the results of ×8 up-sampling considering 5% and 10% noise, respectively.

this section, we provide detailed information on `SuperBench` dataset, such as data hosting, data licensing, data format, and maintenance plans.

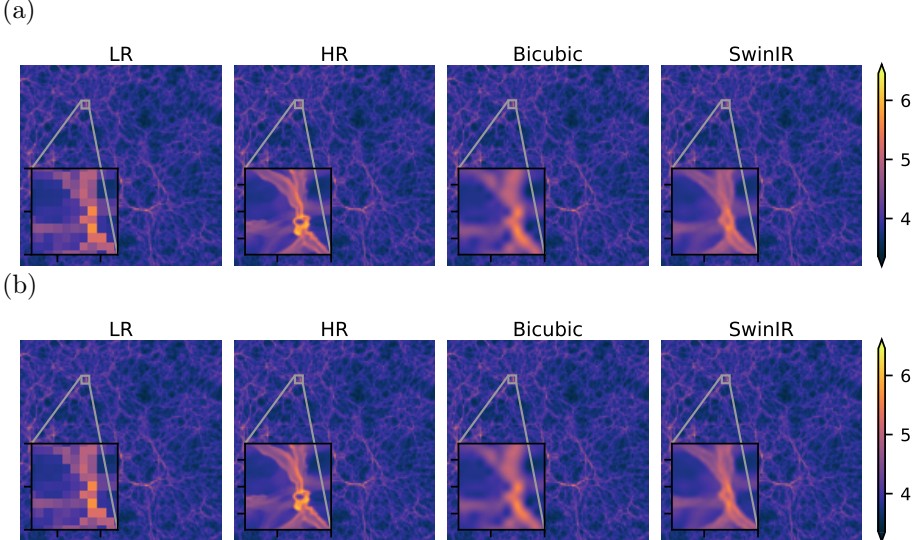

Figure 13: Showcasing baseline SR methods on cosmology data under uniform down-sampling and noise. Here (a) and (b) show the results of ×8 up-sampling considering 5% and 10% noise, respectively.

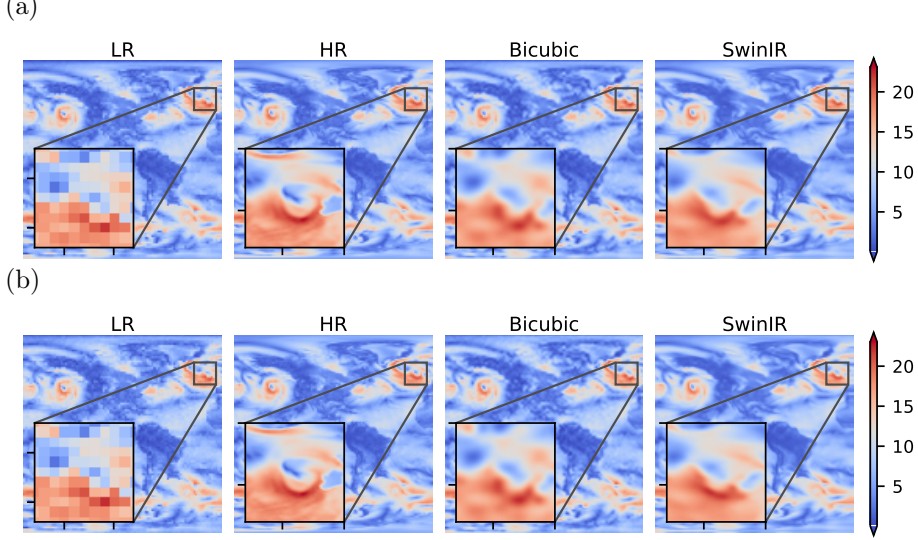

Figure 14: Showcasing baseline SR methods on weather data under uniform down-sampling and noise. Here (a) and (b) show the results of ×8 up-sampling considering 5% and 10% noise, respectively.

## F.1 Data Hosting

SuperBench is hosted on the shared file systems of the National Energy Research Scientific Computing Center (NERSC) platform. The data is publicly available with the following link (https://portal.nersc.gov/project/dasrepo/superbench). Users can download the dataset

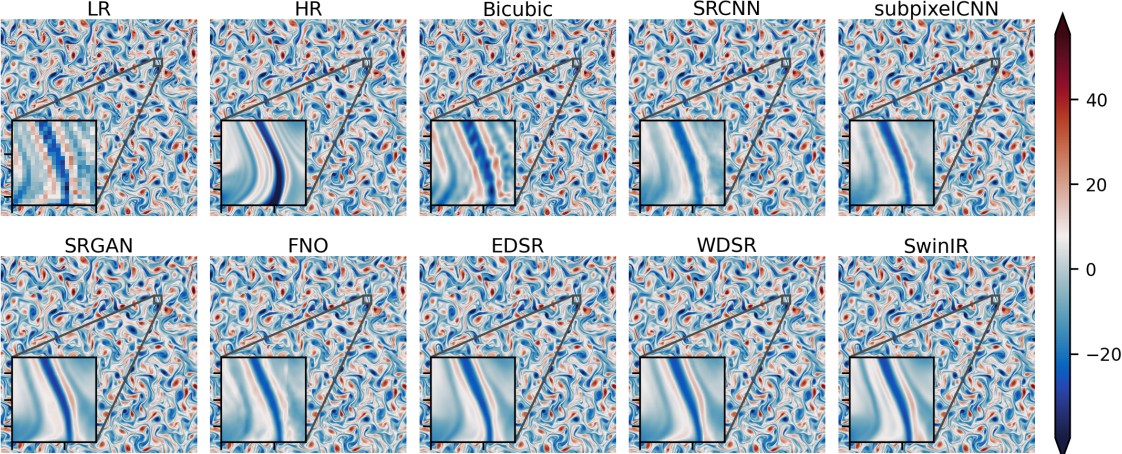

Figure 15: Showcasing baseline SR methods on turbulent fluid flow data ($Re = 16000$) with LR simulation data as inputs. The results are based on ×4 up-sampling.

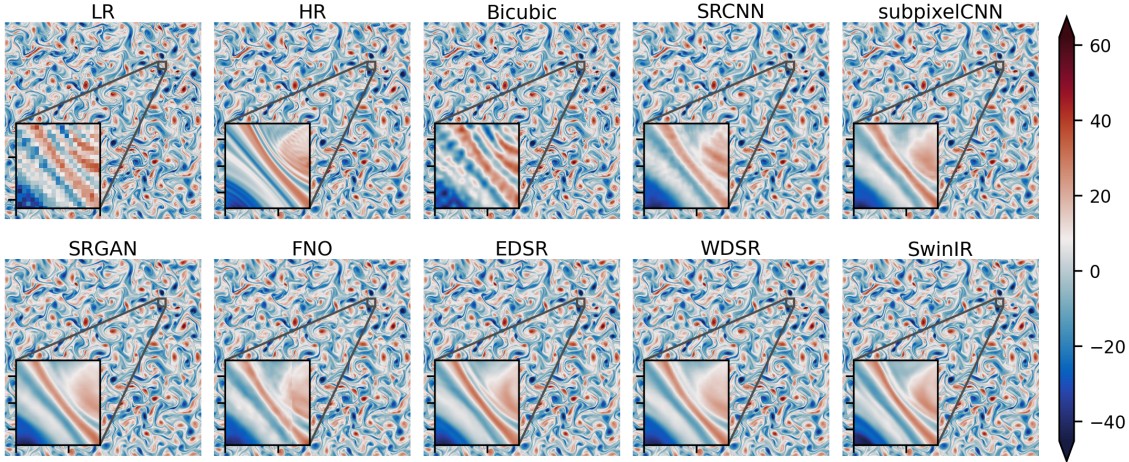

Figure 16: Showcasing baseline SR methods on turbulent fluid flow data ($Re = 32000$) with LR simulation data as inputs. The results are based on ×4 up-sampling.

locally by either clicking on the provided link or using the `wget` command in the terminal. In addition, the code used to process the dataset and run the baseline models is available on our GitHub repository (https://github.com/erichson/SuperBench). We also provide the pre-trained model weights via Google Drive (link).

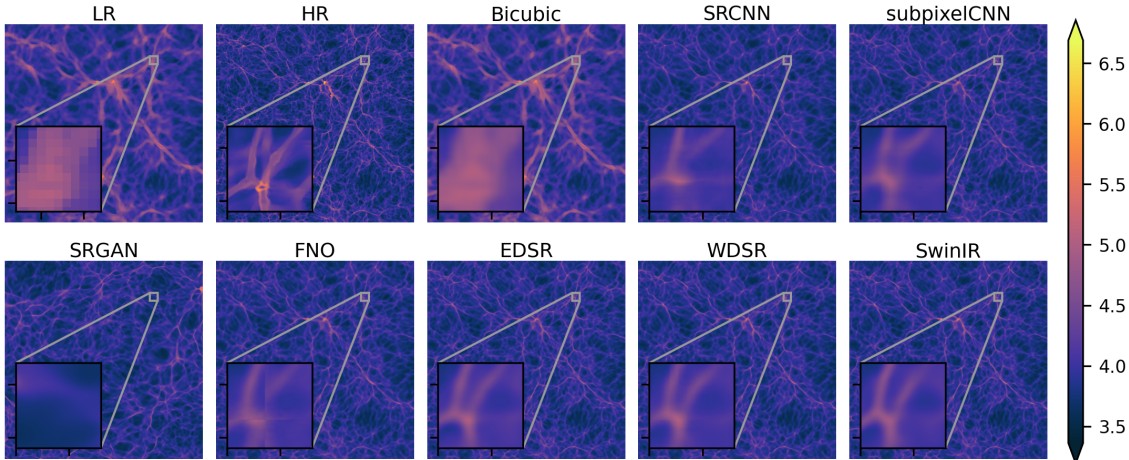

Figure 17: Showcasing baseline SR methods on cosmology data with LR simulation data as inputs. The results are based on ×8 up-sampling.

Table 4: Results for fluid flow data ($Re = 16000$) with bicubic down-sampling.

| Baselines | UF ($\times$) | Interpolation Errors | | | | Extrapolation Errors | | | | # par. |
|---|---|---|---|---|---|---|---|---|---|---|
| | | RFNE | IN | PSNR | SSIM | RFNE | IN | PSNR | SSIM | |
| Bicubic | 8 | 0.12 | 0.25 | 33.18 | 0.91 | 0.12 | 0.19 | 34.68 | 0.92 | 0.00 M |
| FNO | 8 | 0.06 | 0.16 | 29.74 | 0.94 | 0.06 | 0.16 | 29.69 | 0.94 | 4.75 M |
| SRCNN | 8 | 0.13 | 0.28 | 32.29 | 0.91 | 0.12 | 0.23 | 33.62 | 0.92 | 0.07 M |
| subpixelCNN | 8 | 0.04 | 0.09 | 44.83 | 0.99 | 0.04 | 0.07 | 46.23 | 0.99 | 0.34 M |
| SRGAN | 8 | 0.13 | 2.83 | 30.26 | 0.95 | 0.14 | 3.24 | 31.16 | 0.96 | 1.70 M |
| EDSR | 8 | 0.03 | **0.06** | 47.01 | 0.99 | 0.02 | **0.05** | 48.28 | 1.00 | 1.67 M |
| WDSR | 8 | 0.03 | 0.10 | 51.91 | 0.99 | 0.03 | 0.08 | 53.53 | 0.99 | 1.40 M |
| SwinIR | 8 | 0.02 | 0.27 | 51.88 | 1.00 | 0.02 | 0.26 | 52.95 | 1.00 | 12.05 M |
| SwinIR(Phy) ($\lambda_p = 0.001$) | 8 | **0.02** | 0.28 | **54.55** | **1.00** | **0.02** | 0.25 | **55.70** | **1.00** | 12.05 M |
| Bicubic | 16 | 0.22 | 0.37 | 27.57 | 0.82 | 0.21 | 0.31 | 29.03 | 0.85 | 0.00 M |
| FNO | 16 | 0.14 | 0.26 | 22.98 | 0.80 | 0.14 | 0.26 | 22.89 | 0.81 | 4.75 M |
| SRCNN | 16 | 0.23 | 0.43 | 27.08 | 0.83 | 0.23 | 0.36 | 28.50 | 0.85 | 0.07 M |
| subpixelCNN | 16 | 0.08 | 0.15 | 39.30 | 0.93 | 0.08 | 0.12 | 40.53 | 0.95 | 0.67 M |
| SRGAN | 16 | 0.30 | 2.55 | 23.39 | 0.86 | 0.28 | 2.53 | 25.18 | 0.88 | 1.85 M |
| EDSR | 16 | 0.07 | **0.12** | 41.00 | 0.95 | 0.07 | **0.10** | 42.35 | 0.96 | 1.81 M |
| WDSR | 16 | 0.08 | 0.19 | 41.37 | 0.94 | 0.07 | 0.15 | 43.01 | 0.95 | 1.62 M |
| SwinIR | 16 | 0.07 | 0.13 | 41.79 | 0.95 | 0.06 | 0.11 | 43.01 | 0.96 | 12.20 M |
| SwinIR(Phy) ($\lambda_p = 0.001$) | 16 | **0.06** | 0.57 | **44.45** | **0.96** | **0.06** | 0.57 | **45.38** | **0.97** | 12.20 M |

## F.2 Data Licensing

SuperBench is released under an Open Data Commons Attribution License. The fluid flow and cosmology datasets are produced by the authors themselves. The ERA5 dataset provided in SuperBench is available in the public domain.[1] It is free to use for research purposes.

---

1. The corresponding data license can be accessed at: https://cds.climate.copernicus.eu/cdsapp#!/dataset/reanalysis-era5-complete?tab=overview.

Table 5: Results for fluid flow data ($Re = 32000$) with bicubic down-sampling.

| Baselines | UF ($\times$) | Interpolation Errors | | | | Extrapolation Errors | | | | # par. |
|---|---|---|---|---|---|---|---|---|---|---|
| | | RFNE | IN | PSNR | SSIM | RFNE | IN | PSNR | SSIM | |
| Bicubic | 8 | 0.15 | 0.26 | 32.62 | 0.89 | 0.14 | 0.22 | 33.84 | 0.90 | 0.00 M |
| FNO | 8 | 0.08 | 0.18 | 28.63 | 0.92 | 0.08 | 0.18 | 28.46 | 0.92 | 4.75 M |
| SRCNN | 8 | 0.13 | 0.27 | 32.65 | 0.91 | 0.12 | 0.22 | 34.06 | 0.92 | 0.07 M |
| subpixelCNN | 8 | 0.06 | 0.11 | 42.53 | 0.97 | 0.06 | 0.10 | 43.43 | 0.97 | 0.34 M |
| SRGAN | 8 | 0.17 | 2.38 | 29.14 | 0.93 | 0.16 | 2.64 | 29.96 | 0.93 | 1.70 M |
| EDSR | 8 | 0.04 | **0.08** | 45.27 | 0.98 | 0.04 | **0.07** | 46.21 | 0.99 | 1.67 M |
| WDSR | 8 | 0.05 | 0.13 | 48.44 | 0.98 | 0.05 | 0.11 | 49.64 | 0.98 | 1.40 M |
| SwinIR | 8 | 0.04 | 0.52 | 46.56 | 0.99 | 0.04 | 0.50 | 47.52 | 0.85 | 12.05 M |
| SwinIR(Phy) ($\lambda_p = 0.001$) | 8 | **0.03** | 0.42 | **50.53** | 0.99 | **0.03** | 0.42 | **51.30** | **0.99** | 12.05 M |
| Bicubic | 16 | 0.24 | 0.40 | 27.31 | 0.80 | 0.23 | 0.33 | 28.50 | 0.82 | 0.00 M |
| FNO | 16 | 0.16 | 0.28 | 22.38 | 0.77 | 0.16 | 0.28 | 22.21 | 0.77 | 4.75 M |
| SRCNN | 16 | 0.22 | 0.41 | 27.23 | 0.82 | 0.21 | 0.34 | 28.61 | 0.84 | 0.07 M |
| subpixelCNN | 16 | 0.15 | 0.25 | 32.49 | 0.86 | 0.15 | 0.22 | 33.42 | 0.88 | 0.67 M |
| SRGAN | 16 | 0.31 | 2.80 | 23.70 | 0.85 | 0.30 | 2.84 | 24.63 | 0.86 | 1.85 M |
| EDSR | 16 | 0.09 | 0.14 | 39.72 | 0.93 | 0.09 | **0.13** | 40.67 | 0.94 | 1.81 M |
| WDSR | 16 | 0.10 | 0.22 | 39.75 | 0.91 | 0.10 | 0.18 | 40.94 | 0.92 | 1.62 M |
| SwinIR | 16 | 0.09 | **0.14** | 41.31 | 0.93 | 0.08 | 0.13 | 42.20 | 0.94 | 12.20 M |
| SwinIR(Phy) ($\lambda_p = 0.001$) | 16 | **0.08** | 0.76 | **42.24** | **0.94** | **0.08** | 0.75 | **42.87** | **0.94** | 12.20 M |

Table 6: Results for cosmo dataset with bicubic down-sampling.

| Baselines | UF ($\times$) | Interpolation Errors | | | | Extrapolation Errors | | | | # par. |
|---|---|---|---|---|---|---|---|---|---|---|
| | | RFNE | IN | PSNR | SSIM | RFNE | IN | PSNR | SSIM | |
| Bicubic | 8 | 0.36 | 0.65 | 30.02 | 0.77 | 0.36 | 0.65 | 30.21 | 0.77 | 0.00 M |
| FNO | 8 | 0.15 | **0.34** | 29.23 | 0.89 | 0.15 | **0.34** | 28.95 | 0.89 | 4.75 M |
| SRCNN | 8 | 0.36 | 8.68 | 30.27 | 0.79 | 0.36 | 8.24 | 30.09 | 0.78 | 0.06 M |
| subpixelCNN | 8 | 0.16 | 5.72 | 37.39 | 0.95 | 0.16 | 5.47 | 37.31 | 0.95 | 0.30 M |
| SRGAN | 8 | 0.20 | 8.06 | 35.66 | 0.94 | 0.21 | 7.61 | 35.16 | 0.91 | 1.70 M |
| EDSR | 8 | 0.15 | 5.63 | 37.93 | 0.95 | 0.15 | 5.30 | 37.85 | 0.95 | 1.66 M |
| WDSR | 8 | 0.16 | 5.80 | 37.58 | 0.95 | 0.16 | 5.45 | 37.51 | 0.95 | 1.38 M |
| SwinIR | 8 | **0.14** | 5.72 | **38.21** | **0.96** | **0.14** | 5.34 | **38.11** | **0.95** | 12.05 M |
| Bicubic | 16 | 0.58 | 8.86 | 26.12 | 0.55 | 0.59 | 8.54 | 25.93 | 0.55 | 0.00 M |
| FNO | 16 | 0.40 | **0.57** | 20.38 | 0.56 | 0.41 | **0.58** | 20.04 | 0.55 | 4.75 M |
| SRCNN | 16 | 0.59 | 10.03 | 25.93 | 0.57 | 0.60 | 9.63 | 25.78 | 0.56 | 0.06 M |
| subpixelCNN | 16 | 0.39 | 7.68 | 29.61 | 0.71 | 0.39 | 7.40 | 29.38 | 0.71 | 0.52 M |
| SRGAN | 16 | 0.38 | 8.80 | 29.69 | 0.72 | 0.40 | 8.38 | 29.27 | 0.70 | 1.85 M |
| EDSR | 16 | 0.38 | 7.61 | 29.81 | 0.72 | 0.38 | 7.32 | 29.58 | 0.71 | 1.81 M |
| WDSR | 16 | 0.39 | 7.82 | 29.57 | 0.71 | 0.40 | 7.46 | 29.34 | 0.70 | 1.51 M |
| SwinIR | 16 | **0.37** | 7.61 | **29.89** | **0.72** | **0.38** | 7.28 | **29.65** | **0.72** | 12.19 M |

## F.3 Data Format

`SuperBench` consists of seven distinct data files: two NSKT datasets (`nskt_16k` and `nskt_32k`); two NSKT datasets with $\times 4$ LR simulations (`nskt_16k_sim_4` and `nskt_32k_sim_4`); cosmology data (`cosmo`); cosmology data with $\times 8$ LR simulations (`cosmo_sim_8`); and weather data (`climate`). Table 14 shows the variables/channels considered in each dataset. Each data file includes five sub-directories: one training dataset (`train` file), two validation datasets

Table 7: Results for weather dataset with bicubic down-sampling.

| Baselines | UF (×) | Interpolation Errors | | | | Extrapolation Errors | | | | # par. |
|---|---|---|---|---|---|---|---|---|---|---|
| | | RFNE | IN | PSNR | SSIM | RFNE | IN | PSNR | SSIM | |
| Bicubic | 8 | 0.18 | 0.64 | 26.48 | 0.83 | 0.18 | 0.64 | 26.52 | 0.83 | 0.00 M |
| FNO | 8 | 0.13 | **0.47** | 25.33 | 0.79 | 0.13 | **0.46** | 25.37 | 0.79 | 4.75 M |
| SRCNN | 8 | 0.16 | 0.61 | 27.29 | 0.84 | 0.16 | 0.61 | 27.35 | 0.84 | 0.07 M |
| subpixelCNN | 8 | 0.12 | 0.49 | 30.27 | 0.89 | 0.12 | 0.49 | 30.33 | 0.89 | 0.34 M |
| SRGAN | 8 | 0.18 | 2.71 | 25.31 | 0.82 | 0.18 | 2.70 | 25.60 | 0.83 | 1.70 M |
| EDSR | 8 | 0.12 | 0.55 | 30.28 | 0.89 | 0.12 | 0.55 | 30.35 | 0.89 | 1.67 M |
| WDSR | 8 | 0.13 | 0.59 | 29.62 | 0.89 | 0.13 | 0.59 | 29.68 | 0.89 | 1.40 M |
| SwinIR | 8 | **0.11** | 0.48 | **31.21** | **0.90** | **0.11** | 0.48 | **31.28** | **0.90** | 12.05 M |
| Bicubic | 16 | 0.28 | 0.71 | 22.34 | 0.74 | 0.28 | 0.71 | 22.37 | 0.74 | 0.00 M |
| FNO | 16 | 0.22 | 0.58 | 18.86 | 0.68 | 0.21 | 0.67 | 18.98 | 0.68 | 4.75 M |
| SRCNN | 16 | 0.25 | 0.64 | 23.20 | 0.75 | 0.25 | 0.63 | 23.26 | 0.75 | 0.07 M |
| subpixelCNN | 16 | 0.20 | **0.55** | 25.87 | 0.81 | 0.19 | 0.55 | 25.92 | 0.81 | 0.67 M |
| SRGAN | 16 | 0.26 | 2.73 | 22.21 | 0.77 | 0.25 | 2.73 | 22.48 | 0.78 | 1.85 M |
| EDSR | 16 | 0.19 | 0.60 | 25.95 | 0.81 | 0.19 | 0.59 | 26.01 | 0.81 | 1.81 M |
| WDSR | 16 | 0.21 | 0.66 | 24.99 | 0.79 | 0.21 | 0.65 | 25.05 | 0.80 | 1.62 M |
| SwinIR | 16 | **0.18** | 0.56 | **26.40** | **0.82** | **0.18** | **0.55** | **26.43** | **0.82** | 12.20 M |

Table 8: Results of SwinIR for `SuperBench` datasets with uniform down-sampling and 5% noise.

| Datasets | UF (×) | Interpolation Errors | | | | Extrapolation Errors | | | | # par. |
|---|---|---|---|---|---|---|---|---|---|---|
| | | RFNE | IN | PSNR | SSIM | RFNE | IN | PSNR | SSIM | |
| Fluid flow (16k) | 8 | 0.03 | 0.50 | 46.60 | 0.99 | 0.03 | 0.48 | 47.77 | 0.99 | 12.05 M |
| Fluid flow (32k) | 8 | 0.05 | 0.68 | 44.13 | 0.98 | 0.05 | 0.66 | 44.91 | 0.98 | 12.05 M |
| Weather | 8 | 0.12 | 2.14 | 30.01 | 0.89 | 0.12 | 2.21 | 30.04 | 0.89 | 12.05 M |
| Cosmology | 8 | 0.19 | 7.77 | 35.66 | 0.93 | 0.19 | 7.37 | 35.60 | 0.93 | 12.05 M |

Table 9: Results of SwinIR for `SuperBench` datasets with uniform down-sampling and 10% noise.

| Datasets | UF (×) | Interpolation Errors | | | | Extrapolation Errors | | | | # par. |
|---|---|---|---|---|---|---|---|---|---|---|
| | | RFNE | IN | PSNR | SSIM | RFNE | IN | PSNR | SSIM | |
| Fluid flow (16k) | 8 | 0.04 | 0.57 | 42.47 | 0.98 | 0.04 | 0.55 | 43.63 | 0.98 | 12.05 M |
| Fluid flow (32k) | 8 | 0.06 | 0.79 | 41.31 | 0.96 | 0.06 | 0.76 | 42.04 | 0.97 | 12.05 M |
| Weather | 8 | 0.13 | 2.16 | 29.11 | 0.87 | 0.13 | 2.22 | 29.15 | 0.88 | 12.05 M |
| Cosmology | 8 | 0.21 | 7.81 | 35.14 | 0.92 | 0.21 | 7.39 | 35.06 | 0.91 | 12.05 M |

(`valid_1` and `valid_2`) files for validating interpolation and extrapolation performance), and two testing datasets (`test_1` and `test_2`) files for testing interpolation and extrapolation performance). More details about the design of interpolation and extrapolation datasets can be found in Appendix A. The data in each sub-directory is stored in the HDF5 (Folk et al., 2011) binary data format. The code for loading/reading each dataset is provided in the `SuperBench` repository.

Table 10: Results for cosmology with LR simulation data as inputs.

| Baselines | UF (×) | Interpolation Errors | | | | Extrapolation Errors | | | | # par. |
|---|---|---|---|---|---|---|---|---|---|---|
| | | RFNE | IN | PSNR | SSIM | RFNE | IN | PSNR | SSIM | |
| Bicubic | 8 | 1.01 | 8.60 | 21.28 | 0.33 | 1.01 | 8.58 | 21.33 | 0.34 | 0.00 M |
| FNO | 8 | **0.63** | **0.70** | 16.38 | 0.34 | **0.62** | **0.70** | 16.46 | 0.35 | 4.75 M |
| SRCNN | 8 | 0.69 | 9.95 | 24.49 | 0.46 | 0.69 | 10.03 | 24.53 | 0.46 | 0.06 M |
| subpixelCNN | 8 | 0.66 | 9.76 | 24.83 | 0.47 | 0.66 | 9.76 | 24.85 | 0.48 | 0.30 M |
| SRGAN | 8 | 0.64 | 10.01 | **25.08** | **0.49** | 0.65 | 10.00 | **25.10** | **0.50** | 1.70 M |
| EDSR | 8 | 0.67 | 9.79 | 24.80 | 0.48 | 0.67 | 9.87 | 24.82 | 0.48 | 1.66 M |
| WDSR | 8 | 0.66 | 9.72 | 24.85 | 0.48 | 0.66 | 9.79 | 24.88 | 0.48 | 1.38 M |
| SwinIR | 8 | 0.66 | 9.73 | 24.91 | 0.48 | 0.66 | 9.83 | 24.93 | 0.48 | 12.05 M |

Table 11: Results for fluid flow data ($Re = 16000$) with LR simulation data as inputs

| Baselines | UF (×) | Interpolation Errors | | | | Extrapolation Errors | | | | # par. |
|---|---|---|---|---|---|---|---|---|---|---|
| | | RFNE | IN | PSNR | SSIM | RFNE | IN | PSNR | SSIM | |
| Bicubic | 4 | 0.30 | 3.26 | 24.17 | 0.75 | 0.31 | 4.01 | 23.85 | 0.74 | 0.00 M |
| FNO | 4 | 0.28 | **0.52** | 17.91 | 0.64 | 0.29 | **0.52** | 17.86 | 0.63 | 4.75 M |
| SRCNN | 4 | 0.25 | 3.05 | 25.43 | 0.79 | 0.26 | 3.22 | 25.03 | 0.78 | 0.07 M |
| subpixelCNN | 4 | 0.23 | 2.91 | 26.10 | 0.80 | 0.25 | 3.31 | 25.58 | 0.79 | 0.26 M |
| SRGAN | 4 | 0.25 | 3.21 | 25.52 | 0.81 | 0.26 | 3.70 | 24.97 | 0.79 | 1.55 M |
| EDSR | 4 | 0.22 | 2.87 | **26.70** | 0.82 | 0.24 | 3.42 | **25.82** | 0.80 | 1.52 M |
| WDSR | 4 | 0.23 | 2.97 | 26.18 | 0.81 | 0.24 | 3.33 | 25.62 | 0.80 | 1.35 M |
| SwinIR | 4 | **0.22** | 2.78 | 26.63 | **0.82** | **0.24** | 3.32 | 25.61 | **0.80** | 11.90 M |
| SwinIR(Phy) ($\lambda_p = 0.001$) | 4 | 0.23 | 2.80 | 26.24 | 0.81 | 0.25 | 3.27 | 25.32 | 0.79 | 11.90 M |

Table 12: Results for fluid flow data ($Re = 32000$) with LR simulation data as inputs

| Baselines | UF (×) | Interpolation Errors | | | | Extrapolation Errors | | | | # par. |
|---|---|---|---|---|---|---|---|---|---|---|
| | | RFNE | IN | PSNR | SSIM | RFNE | IN | PSNR | SSIM | |
| Bicubic | 4 | 0.33 | 3.80 | 23.77 | 0.73 | 0.34 | 3.84 | 23.18 | 0.72 | 0.00 M |
| FNO | 4 | 0.31 | **0.55** | 13.26 | 0.59 | 0.32 | **0.58** | 12.86 | 0.58 | 4.75 M |
| SRCNN | 4 | 0.27 | 3.36 | 24.99 | 0.77 | 0.28 | 3.64 | 24.35 | 0.75 | 0.07 M |
| subpixelCNN | 4 | 0.26 | 3.34 | 25.65 | 0.78 | 0.27 | 3.75 | 24.80 | 0.77 | 0.26 M |
| SRGAN | 4 | 0.28 | 3.71 | 24.92 | 0.78 | 0.29 | 3.83 | 24.10 | 0.77 | 1.55 M |
| EDSR | 4 | 0.25 | 3.29 | 25.88 | 0.79 | 0.27 | 3.71 | 24.77 | 0.77 | 1.52 M |
| WDSR | 4 | 0.25 | 3.34 | 25.84 | 0.79 | 0.27 | 3.72 | **24.94** | 0.78 | 1.35 M |
| SwinIR | 4 | **0.24** | 3.30 | 26.24 | 0.80 | 0.27 | 3.58 | 24.91 | 0.78 | 11.90 M |
| SwinIR(Phy) ($\lambda_p = 0.001$) | 4 | 0.25 | 3.27 | 26.27 | **0.80** | **0.27** | 3.62 | 24.93 | **0.78** | 11.90 M |

## F.4 Maintenance Plan

`SuperBench` will be maintained by LBNL. Any encountered issues will be promptly addressed and updated in the provided data and GitHub repository links. Moreover, as highlighted in the Conclusion (Section 6), `SuperBench` offers an extendable framework to include new datasets and baseline models. We encourage contributions from the SciML community and envision `SuperBench` as a collaborative platform. Future versions of `SuperBench` dataset will be released.

Table 13: Results of physics errors on fluid flow data.

| Methods | UF (×) | Down-sampling | Noise (%) | $Re = 16000$ Interp. | Extrap. | $Re = 32000$ Interp. | Extrap. | # par. |
|---|---|---|---|---|---|---|---|---|
| Bicubic | 8 | Bicubic | 0 | 0.92 | 0.88 | 1.00 | 0.96 | 0.00 M |
| SRCNN | 8 | Bicubic | 0 | 4.72 | 4.16 | 2.87 | 2.53 | 0.07 M |
| FNO | 8 | Bicubic | 0 | 12.18 | 11.70 | 11.61 | 11.82 | 4.75 M |
| subpixelCNN | 8 | Bicubic | 0 | 0.68 | 0.61 | 0.96 | 0.88 | 0.34 M |
| SRGAN | 8 | Bicubic | 0 | 5502.12 | 5048.78 | 8540.48 | 8077.06 | 1.70 M |
| EDSR | 8 | Bicubic | 0 | 0.19 | 0.16 | 0.30 | 0.27 | 1.67 M |
| WDSR | 8 | Bicubic | 0 | 0.61 | 0.54 | 0.93 | 0.85 | 1.40 M |
| SwinIR | 8 | Bicubic | 0 | 0.09 | 0.07 | 0.04 | 0.04 | 12.05 M |
| SwinIR ($\lambda_p = 0.001$) | 8 | Bicubic | 0 | **0.02** | **0.02** | **0.02** | **0.02** | 12.05 M |
| Bicubic | 16 | Bicubic | 0 | 0.82 | 0.79 | 0.88 | 0.84 | 0.00 M |
| SRCNN | 16 | Bicubic | 0 | 3.31 | 3.04 | 3.50 | 3.22 | 0.07 M |
| FNO | 16 | Bicubic | 0 | 25.57 | 25.55 | 20.33 | 20.21 | 4.75 M |
| subpixelCNN | 16 | Bicubic | 0 | 0.70 | 0.64 | 42.85 | 40.11 | 0.67 M |
| SRGAN | 16 | Bicubic | 0 | 21688.79 | 20672.48 | 38924.85 | 38117.07 | 1.85 M |
| EDSR | 16 | Bicubic | 0 | 0.49 | 0.44 | 0.36 | 0.32 | 1.81 M |
| WDSR | 16 | Bicubic | 0 | 1.79 | 1.66 | 1.45 | 1.32 | 1.62 M |
| SwinIR | 16 | Bicubic | 0 | 0.50 | 0.43 | 0.30 | 0.26 | 12.20 M |
| SwinIR ($\lambda_p = 0.001$) | 16 | Bicubic | 0 | **0.05** | **0.04** | **0.05** | **0.04** | 12.20 M |
| Bicubic | 4 | LR Simulation | 0 | 1.99 | 2.01 | 2.51 | 2.54 | 0.00 M |
| SRCNN | 4 | LR Simulation | 0 | 11.56 | 11.78 | 13.77 | 14.08 | 0.07 M |
| FNO | 4 | LR Simulation | 0 | 75.15 | 76.20 | 86.06 | 89.85 | 4.75 M |
| subpixelCNN | 4 | LR Simulation | 0 | 8.85 | 8.76 | 11.35 | 11.88 | 0.26 M |
| SRGAN | 4 | LR Simulation | 0 | 22001.4 | 22155.21 | 16404.89 | 16522.42 | 1.55 M |
| EDSR | 4 | LR Simulation | 0 | 4.34 | 4.35 | 4.34 | 4.21 | 1.52 M |
| WDSR | 4 | LR Simulation | 0 | 4.66 | 4.78 | 6.11 | 6.22 | 1.35 M |
| SwinIR | 4 | LR Simulation | 0 | 1.18 | 1.16 | 1.35 | 1.30 | 11.90 M |
| SwinIR ($\lambda_p = 0.001$) | 4 | LR Simulation | 0 | **0.29** | **0.28** | **0.30** | **0.29** | 11.90 M |
| SwinIR | 8 | Noisy Uniform | 5 | 0.14 | 0.12 | 0.14 | 0.12 | 12.05 M |
| SwinIR | 8 | Noisy Uniform | 10 | 0.18 | 0.15 | 0.18 | 0.15 | 12.05 M |

Table 14: Summary of variables in `SuperBench` datasets.

| Data name | Variables/Channels (in order) |
|---|---|
| `nskt_16k` | velocity (x), velocity (y), and vorticity |
| `nskt_16k_sim_4` | velocity (x), velocity (y), and vorticity |
| `nskt_32k` | velocity (x), velocity (y), and vorticity |
| `nskt_32k_sim_4` | velocity (x), velocity (y), and vorticity |
| `cosmo` | temperature and baryon density |
| `cosmo_sim_8` | temperature and baryon density |
| `climate` | KE at 10m from surface, temperature at 2m from surface, and total column water vapor |

# Appendix G. Guidelines for Responsible Use

To ensure the responsible use of our `SuperBench` datasets, we propose the following guidelines:

**Fair and transparent comparison.** Users are encouraged to report experimental results clearly and consistently, following standardized metrics provided in Table 2. We also suggest

to present details about hyperparameters, training protocols, and preprocessing steps to enable reproducibility.

**Ethical principles.** The datasets should only be used for research and educational purposes. Commercial or malicious applications are discouraged. Users must respect intellectual property rights and appropriately credit the dataset creators and contributors as detailed in the licensing terms.

**Domain context.** Users are encouraged to use the domain-specific metrics included in SuperBench, such as continuity errors and energy spectrum evaluation for fluid dynamics if applicable. SR methods, if not properly validated, could introduce artifacts or inaccuracies that may lead to misleading scientific conclusions.

**Community contributions.** SuperBench is designed to be extensible, and we welcome the addition of new datasets or evaluation metrics. Contributors should ensure these additions align with the existing structure and standards. Users are encouraged to share feedback and improvements for inclusion in future updates.

## Appendix H. Datasheet

### H.1 Motivation

- For what purpose was the dataset created? *SuperBench serves as a benchmark dataset for evaluating spatial SR methods in scientific applications.*

- Who created this dataset (e.g., which team, research group) and on behalf of which entity (e.g., company, institution, organization)? *This dataset is a collaborative effort involving a diverse team from different institutes, including Lawrence Berkeley National Lab (LBNL), University of California at Berkeley, International Computer Science Institute (ICSI), and the University of Tennessee.*

- Who funded the creation of the dataset? *The main funding body is the U.S. Department of Energy, Office of Science, Office of Advanced Scientific Computing Research, Scientific Discovery through Advanced Computing (SciDAC) program. Other funding is from the National Energy Research Scientific Computing Center (NERSC) at Lawrence Berkeley National Laboratory.*

### H.2 Composition

- What do the instances that comprise the dataset represent (e.g., documents, photos, people, countries)? *Each instance contains a pair of LR input and HR output. The inputs and outputs are snapshots of various scientific data.*

- How many instances are there in total (of each type, if appropriate)? *The LR and HR pairs include 11455 instances for all these datasets in SuperBench.*

- Does the dataset contain all possible instances or is it a sample (not necessarily random) of instances from a larger set? *No, this dataset is a subset of larger sets.*

- What data does each instance consist of? "Raw" data (e.g., unprocessed text or images) or features? *Each instance consists of LR downgraded data and HR simulation data.*

- Is there a label or target associated with each instance? *Yes, each instance includes a target HR data.*

- Is any information missing from individual instances? *No.*

- Are relationships between individual instances made explicit (e.g., users' movie ratings, social network links)? *No.*

- Are there recommended data splits (e.g., training, development/validation, testing)? *The data split has already been done in the stage of preprocessing. We use "train", "valid1", "valid2", "test1", and "test2" to represent the training set, in-distribution validation set, out-of-distribution validation set, in-distribution testing set, out-of-distribution testing set. The detailed data splitting is presented in Table 1.*

- Are there any errors, sources of noise, or redundancies in the dataset? *No.*

- Is the dataset self-contained, or does it link to or otherwise rely on external resources (e.g., websites, tweets, other datasets)? *The weather data is from the original ERA5 dataset. The fluid and cosmology datasets are self-contained.*

- Does the dataset contain data that might be considered confidential (e.g., data that is protected by legal privilege or by doctor-patient confidentiality, data that includes the content of individuals non-public communications)? *No.*

- Does the dataset contain data that, if viewed directly, might be offensive, insulting, threatening, or might otherwise cause anxiety? *No.*

- Does the dataset relate to people? *No.*

- Does the dataset identify any subpopulations (e.g., by age, gender)? *No.*

- Is it possible to identify individuals (i.e., one or more natural persons), either directly or indirectly (i.e., in combination with other data) from the dataset? *No.*

- Does the dataset contain data that might be considered sensitive in any way (e.g., data that reveals racial or ethnic origins, sexual orientations, religious beliefs, political opinions or union memberships, or locations; financial or health data; biometric or genetic data; forms of government identification, such as social security numbers; criminal history)? *No.*

### H.3 Collection Process

- How was the data associated with each instance acquired? *The fluid data associated with each instance is acquired from Direct Numerical Simulation (DNS). The cosmology data is simulated using the `Nyx` code. The weather data is subsampled from the ERA5 dataset. More detailed information can be found in Section 3.1.*

- What mechanisms or procedures were used to collect the data (e.g., hardware apparatus or sensor, manual human curation, software program, software API)? *The fluid and*

*cosmology data are simulated using multiple Nvidia Tesla A100 GPU nodes in Permultter, which is a supercomputer at LBNL.*

- Who was involved in the data collection process (e.g., students, crowdworkers, contractors) and how were they compensated (e.g., how much were crowdworkers paid)? *Postdoc fellows and scientists from LBNL as well as a professor from the University of Tennessee were involved in the data collection process.*

- Does the dataset relate to people? *No.*

- Did you collect the data from the individuals in question directly, or obtain it via third parties or other sources (e.g., websites)? *We collect fluid and cosmology data from numerical simulations. The weather data is from the original ERA5 dataset.*

### H.4 Preprocessing/cleaning/labeling

- Was any preprocessing/cleaning/labeling of the data done (e.g., discretization or bucketing, tokenization, part-of-speech tagging, SIFT feature extraction, removal of instances, processing of missing values)? *Yes. The inputs and outputs have been labeled as LR and HR pairs.*

- Was the "raw" data saved in addition to the preprocessed/cleaned/labeled data (e.g., to support unanticipated future uses)? *Yes. The "raw" data is saved in Perlmutter for future use.*

- Is the software used to preprocess/clean/label the instances available? *Yes. The code is saved in Perlmutter for future use.*

### H.5 Uses

- Has the dataset been used for any tasks already? *No.*

- Is there a repository that links to any or all papers or systems that use the dataset? *Yes. The repository is provided at [https://github.com/erichson/SuperBench](https://github.com/erichson/SuperBench).*

- What (other) tasks could the dataset be used for? *It can also be used for other image-related tasks, such as watermarking. The weather dataset is also applicable for spatiotemporal forecasting tasks.*

- Are there tasks for which the dataset should not be used? *The fluid and cosmology data are unsuitable for spatiotemporal forecasting.*

### H.6 Distribution

- Will the dataset be distributed to third parties outside of the entity (e.g., company, institution, organization) on behalf of which the dataset was created? *Yes, the dataset is available to the public.*

- How will the dataset be distributed (e.g., tarball on website, API, GitHub) *The dataset is distributed through the shared file systems of the National Energy Research Scientific*

*Computing Center (NERSC) platform. The code is distributed through the GitHub repository.*

- Will the dataset be distributed under a copyright or other intellectual property (IP) license, and/or under applicable terms of use (ToU)? *Yes, the data is distributed under an Open Data Commons Attribution License.*

- Have any third parties imposed IP-based or other restrictions on the data associated with the instances? *No.*

- Do any export controls or other regulatory restrictions apply to the dataset or to individual instances? *No.*

## H.7 Maintenance

- Who will be supporting/hosting/maintaining the dataset? *LBNL will be supporting/hosting/maintaining the dataset.*

- How can the owner/curator/manager of the dataset be contacted (e.g., email address)? The owner/curator/manager of the dataset be contacted with: `pren@lbl.gov` (Pu Ren) and `erichson@lbl.gov` (N. Benjamin Erichson).

- Is there an erratum? *No. If there is any error, we will update the associated data or code immediately and post it on our GitHub repository.*

- Will the dataset be updated (e.g., to correct labeling errors, add new instances, delete instances)? *Yes. We will update the associated data or code using the provided data and GitHub repository links.*

- If the dataset relates to people, are there applicable limits on the retention of the data associated with the instances (e.g., were individuals in question told that their data would be retained for a fixed period of time and then deleted)? *N/A.*

- Will older versions of the dataset continue to be supported/hosted/maintained? *Yes.*

- If others want to extend/augment/build on/contribute to the dataset, is there a mechanism for them to do so? *Yes. `SuperBench` offers an extendable framework to include new datasets and baseline models. The researchers may consider opening an issue on the GitHub repository and providing a link to your datasets with data details.*

