# OpenReview forum: "SuperBench: A Super-Resolution Benchmark Dataset for Scientific Machine Learning"
_DMLR — Accepted by DMLR_

### Review · Reviewer_26cM · 2024-11-29

**Recommendation:** 4
**Confidence:** 2

**Summary Of Contributions:**

In this paper, the authors introduce a benchmark dataset featuring high-resolution datasets of fluid flows, cosmology, and weather. This paper focuses on validating spatial SR performance from data-centric and physics-preserved perspectives, while assessing robustness to data degradation tasks.

**Strengths:**

Please find Strengths And Weaknesses.

**Audience:**

Yes

**Claims And Evidence:**

Yes.

**Datasets And Benchmarks:**

Yes.

**Extended Submissions:**

No.

**Limitations:**

Please find Strengths And Weaknesses.

**Requested Changes:**

Please find Strengths And Weaknesses.

**Strengths And Weaknesses:**

Thanks for introducing the dataset. This paper is well organized and easy to follow. This paper introduces a new high-resolution benchmark dataset for spatiotemporal systems, which is meaningful for the development of scientific machine learning.

---

### Review · Reviewer_65Ei · 2024-12-02

**Recommendation:** 4
**Confidence:** 2

**Summary Of Contributions:**

This manuscript proposed SuperBench, a benchmark dataset that is used for assessing SR methods for research settings. The proposed contributions are:
1. A large benchmark dataset that includes 4 HR simulations, with up to 2k*2k dimension.
2. Implemented and evaluated a range of degradation functions for scientific data, and included existing SR methods for analysis
3. Baseline evaluations and various metrics, including physics metrics.

**Strengths:**

Contribution: provided a standardized benchmark datasets for evaluating SR methods.

Quality: Reasonable methodology, various evaluation metrics, and description of data collection and preprocessing

Reproducibility: Publicly available code and data, with documentation and maintenance plans

Clarity: Relatively smooth text, supplemented with experimental results and appendix for additional information.

**Audience:**

Yes

**Broader Impact Concerns:**

The paper addresses the broader impact considerations, with documentation, discussion of limitations, maintenance, and source availability.

There seems to be no major ethical concerns, thought relatively discussion is somewhat limited

**Claims And Evidence:**

The claims are supported with evidences.

**Datasets And Benchmarks:**

The paper appears to have the required details, but ethical considerations around dataset usage are relatively light

**Extended Submissions:**

This does not look like a previously published work

**Limitations:**

Focused on spatial SR

Potential misuse of the benchmark

More baseline comparisons

**Requested Changes:**

Clear guidelines for responsible use of the datasets, ethical principles, fair comparison, etc.

Suggested but not required:

The manuscript could benefit if the author could expand some discussion on negative uses, pitfalls; and maybe add more baseline comparisons

**Strengths And Weaknesses:**

Strengths:

1. SR benchmarking
2. High-quality, diverse SR datasets
3. A evaluation framework with various metrics
4. Reproducibility with public code/data
5. Clear documentation and maintenance plans


Weaknesses:

1. Limited to spatial SR only
2. Could benefit from more discussion of potential negative uses, and on related work, if possible
3. Baseline comparisons could be expanded

---

### Review · Reviewer_a8iv · 2024-12-17

**Recommendation:** 4
**Confidence:** 2

**Summary Of Contributions:**

This paper introduces SuperBench, a new benchmark dataset specifically designed for evaluating super-resolution methods in scientific domains. The authors construct datasets from three scientific fields: fluid flows, cosmology, and weather. The benchmark provides multiple degradation scenarios including bicubic downsampling, noisy uniform downsampling, and direct low-resolution simulations. The authors also evaluate several super-resolution methods. One key finding is that they struggle to preserve important physical properties.

**Strengths:**

1. It is very meaningful to have a scientific super-resolution benchmark.
2. The benchmark has included the fluid flows, cosmology, and weather data, which seem underrepresented in traditional SR benchmarks.
3. The provided data has high resolution (up to 2048×2048).
4. The work clearly identifies current limitations of SR methods in scientific applications, which helps guide future research.

**Audience:**

Yes

**Claims And Evidence:**

Yes

**Datasets And Benchmarks:**

Yes

**Extended Submissions:**

N/A

**Limitations:**

**Limited Consideration of 3D Data**
- The benchmark only considers 2D fluid flows, while real-world scientific applications often deal with 3D data.
- For cosmology data, while using 2D slices from 3D simulations is reasonable, this approach cannot guarantee physical consistency between slices and may miss important 3D physical properties.
The authors should explain why they didn't explore 3D SR directly, as this would better represent real scientific challenges.

**Dataset Design Choices**
The authors claim that "the spatial resolution of the datasets is carefully chosen" to enable training without multi-GPU computing. However, they don't specify their selection criteria or provide any analysis of the computational requirements.

**Evaluation**
The current evaluation metrics are quite basic. While the authors mention that researchers can incorporate domain-specific metrics, the benchmark itself should provide more comprehensive domain-motivated evaluation criteria. This would help ML researchers better understand and address domain-specific challenges in their model design.

**Requested Changes:**

N/A

**Strengths And Weaknesses:**

In general, this paper provides the super-resolution dataset for scientific domains, which is very meaningful. The dataset and evaluation framework could greatly encourage both machine learning researchers and domain scientists to tackle the unique challenges in scientific super-resolution. However, it only includes three domains (fluid flows, cosmology, and weather) with limited variants, which may not fully represent the diverse challenges across scientific fields.